# A click chemistry amplified nanopore assay for ultrasensitive quantification of HIV-1 p24 antigen in clinical samples

Xiaojun Wei [1,2], Xiaoqin Wang[2], Zehui Zhang [1], Yuanyuan Luo[3], Zixin Wang [3], Wen Xiong[2], Piyush K. Jain [4,5,6], John R. Monnier[2], Hui Wang [3], Tony Y. Hu [7,8], Chuanbing Tang [3], Helmut Albrecht[9,10] & Chang Liu [1,2] ✉

Despite major advances in HIV testing, ultrasensitive detection of early infection remains challenging, especially for the viral capsid protein p24, which is an early virological biomarker of HIV-1 infection. Here, To improve p24 detection in patients missed by immunological tests that dominate the diagnostics market, we show a click chemistry amplified nanopore (CAN) assay for ultrasensitive quantitative detection. This strategy achieves a 20.8 fM (0.5 pg/ml) limit of detection for HIV-1 p24 antigen in human serum, demonstrating 20-100-fold higher analytical sensitivity than nanocluster-based immunoassays and clinically used enzyme-linked immunosorbent assay, respectively. Clinical validation of the CAN assay in a pilot cohort shows p24 quantification at ultra-low concentration range and correlation with CD4 count and viral load. We believe that this strategy can improve the utility of p24 antigen in detecting early infection and monitoring HIV progression and treatment efficacy, and also can be readily modified to detect other infectious diseases.

Despite the decrease of Human Immunodeficiency Virus (HIV) incidence, the overall infected population continues to grow. It is estimated that 20% of new HIV infections are due to transmission from unaware infected individuals[1]. Hence, early detection of HIV and anti-retroviral therapies monitoring are particularly important for improving patient outcomes and lowering transmission rates[2,3]. To this end, extending the detection window and the monitoring capacity of HIV biomarker assays by achieving a lower limit of detection (LOD) and accurate quantification is urgently required[4,5]. Circulating RNA and antigen p24 have long been recognized as two significant HIV-1 virological biomarkers, especially in early attempts to close the window period of detection[6]. Favored by the development of highly efficient amplification techniques such as polymerase chain reaction (PCR), procedures for quantifying HIV-1 viral nucleic acids in plasma has become a major endpoint parameter for clinical evaluations of diagnosis and antiretroviral therapies. That said, PCR methods are not without their limitations. Apart from the time consumption and expensive costs for purification and amplification usually required, such tests will miss the point that p24 antigen, a 24-25 kDa protein encoded by the gag gene, can correlate significantly with host immune parameters, especially for patients in whom HIV RNA is difficult to detect[7]. Additionally, some early studies investigating patients soon

[1]Biomedical Engineering Program, University of South Carolina, Columbia, SC 29208, USA. [2]Department of Chemical Engineering, University of South Carolina, Columbia, SC 29208, USA. [3]Department of Chemistry and Biochemistry, University of South Carolina, Columbia, SC 29208, USA. [4]Department of Chemical Engineering, University of Florida, Gainesville, FL 32611, USA. [5]Department of Molecular Genetics and Microbiology, University of Florida, Gainesville, FL 32610, USA. [6]UF Health Cancer Center, University of Florida, Gainesville, FL 32608, USA. [7]Center for Cellular and Molecular Diagnostics, Tulane University School of Medicine, New Orleans, LA 70112, USA. [8]Department of Biochemistry and Molecular Biology, Tulane University School of Medicine, New Orleans, LA 70112, USA. [9]Department of Internal Medicine, School of Medicine, University of South Carolina, Columbia, SC 29209, USA. [10]Center of Infectious Diseases Research and Policy, Prisma Health, Columbia, SC 29203, USA. ✉e-mail: changliu@cec.sc.edu

after seroconversion indeed found that p24 detectability was a stronger predictor of progression to acquired immune deficiency syndrome (AIDS) than was RNA[3,8,9]. Although PCR technology has achieved low-concentration detection of viral RNA, there is no evidence showing that RNA appears ahead of p24 before seroconversion. The major challenge for using p24 as a biomarker is that proteins cannot be amplified like nucleic acids[10], leading to the widely-held belief that p24 tests are relatively insensitive and therefore have a limited utility in clinical practice[11], which is in fact a technological issue[12,13].

Emerging approaches for p24 detection based on different combinations of nanoparticles/beads, enzymes, and antibodies have been reported[10,14–16]. Although conceptual assays have exceeded the practical LOD for p24 in patients (3-4 pg/mL, 0.13-0.17 pM)[17], many of these are early-stage experimental studies and have not yet made a successful translation through clinical studies. At present, nanomaterial-based fluorescence assay and enzyme-linked immunosorbent assay (ELISA) are still commonly used to detect p24 in human serum. The sensitivity of these assays in clinical practices, however, are far from satisfactory[2,18]. Further improvements are needed to realize early detection of p24 at the low concentration range in clinical specimens.

The nanopore technology with single-molecule level sensitivity has witnessed escalating interests both in research and commercial fields[19–23]. By measurement and statistical analysis of ionic current blockades produced by translocations of individual target analytes through a single nanopore under an applied electrical potential, the concentration of an analyte can be obtained via the frequency of blockade events[24–27]. In general, concentration of analyte, such as DNA, should be at least at nanomolar level to capture a statistically significant number of translocation events in a reasonable measurement time[28]. Previous works have applied nanopore to HIV-related research including measuring HIV-1 protease activity[29], detecting HIV-1 DNAs[30], etc. However, due to its stochastic nature, it is difficult to specifically sense ultra-low abundance biomarkers mixed with interferent molecules in complex clinical specimens using a nanopore without any recognition receptors[31]. Therefore, sensitivity of direct detection of antigens and antibodies through electrophoresis-based nanopore sensing is limited, especially using biological nanopores with limited operating pH and fixed pore diameters[32,33]. Highly charged DNA

molecule can effectively pass through biological nanopores[34], thus have been used as driving probes to enhance sensitivity for protein detection[35,36]. Furthermore, quantitative chemical modifications to DNA probes with concentration dependent probe yield and highly characteristic current blockade events were utilized in nanopore sensing to improve sensitivity and recognizability of the desired signals[37]. For example, by incorporating a sandwich assay involving copper oxide nanoparticles, a host-guest modified DNA probe catalyzed by assay released Cu ions was used to derive concentrations of cancer biomarkers[38]. Our group recently used two DNA structures with different modifications as detection reporters for multiplex quantification of immunoglobulin M and G antibodies against the nucleocapsid protein of SARS-CoV-2 in serum specimens from early stage COVID-19 patients[39].

The above-mentioned studies have made great progress in several aspects in developing DNA probe assisted nanopore sensing for various biomarkers with high selectivity and sensitivity. However, many challenges remain ahead towards the goal of a universal biosensing platform. First, a robust amplification strategy is required to further improve the detection sensitivity and achieve a lower LOD for accurate early detection. Second, relationships between DNA probes, the biomarker, the catalyst, and the host-guest structure in the modification process, as well as the distribution of their respective nanopore signals remain ill-defined, thus hinders further improvements on sensing accuracy and reproducibility. Third, a rigorous clinical performance evaluation is currently lacking, which is of utmost importance in the last mile from bench to bedside.

Herein, we report an ultrasensitive Click chemistry Amplified Nanopore (CAN) assay engineered for HIV-1 p24 antigen quantification in human serum. The principle of the CAN assay is based on the combination of an amplified sandwich assay and nanopore sensing, which involves several steps (Fig. 1). Comparing to our previous DNA-assisted nanopore sensor for SARS-CoV-2 antibodies[39], in which detection solely depends on separation by immunosorbent beads and conversion to DNA probes preloaded on gold nanoparticles together with detection antibodies, the present CAN assay has two build-in amplifications by catalytic click reaction and biotin-avidin binding that contribute to a lower LOD to allow early detection of antigens in sera.

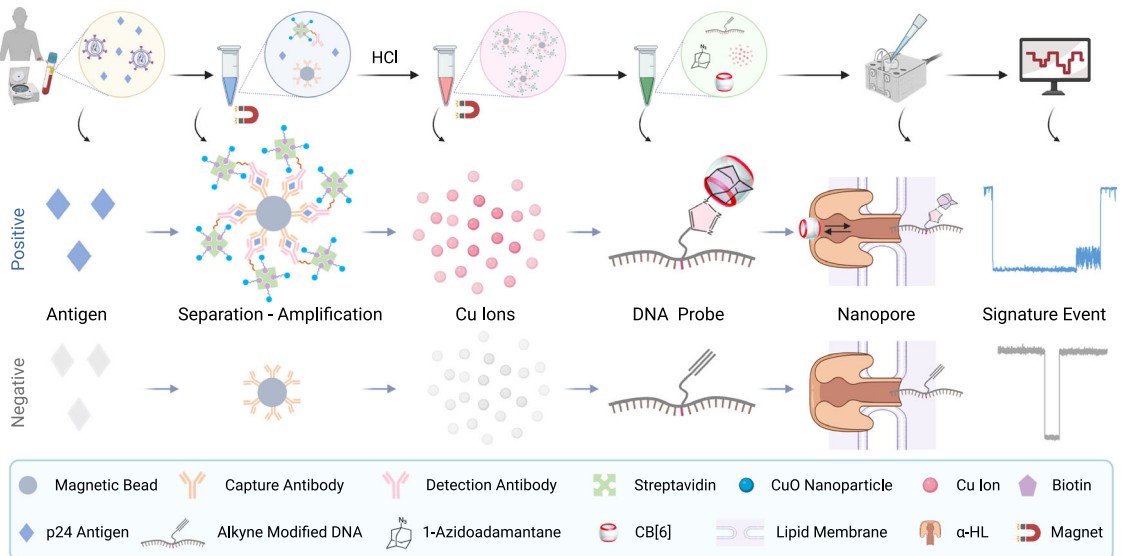

**Fig. 1 | Illustration of the click chemistry amplified nanopore (CAN) assay workflow for quantification of HIV-1 p24 antigen in human serum.** (1) blood samples were collected into serum separator tubes, and sera were collected after centrifugation and clot formation; (2) serum samples were incubated with capture antibody-modified magnetic beads (MBs) and detection antibody-modified copper oxide nanoparticles (CuONPs) to form a sandwich structure between MBs, enriched p24 antigens and CuONPs; (3) sandwich complexes were magnetically separated, and Cu⁺ ions were released from them under acidic conditions; (4) DNA probes were formed by a Cu⁺ ion catalyzed click reaction and a modification with a pair of host-guest molecules; (5) finally, DNA probes were collected and subjected to single-channel recordings using an α-HL nanopore for p24 antigen quantification. The illustration is not drawn to scale. (Created with BioRender.com).

To ensure high specificity, the DNA probe is designed to induce a unique translocation signal that is clearly different from signals of other molecules in the assay system. Mechanisms for enhancing assay sensitivity and specificity were comprehensively characterized and optimized. As a result, this assay achieved a LOD of 20.8 fM (0.5 pg/mL) for p24 detection in human serum, demonstrating 40-100-fold higher analytical sensitivity than a fluorescent copper nanoclusters (CuNCs) assay (Supplementary Fig. 1) and 20-fold higher analytical sensitivity than a clinically used benchmark ELISA, respectively. The CAN assay was validated in correlation with CD4 count and viral load using clinical specimens from a HIV-1 pilot cohort. Comparison to a commercialized fourth-generation ELISA-based HIV-1 test demonstrated superior p24 detection sensitivity and quantification capacity of the CAN assay in patients with lower viral loads, which indicate potential utility in early diagnosis of acute HIV-1 infection and viral load monitoring.

## Results

### Nanopore Characterization and Translocation Behaviors of DNA Probes

The abovementioned modifications to the DNA probes can endow enhanced sensitivity and specificity for nanopore sensing. For characterization, DNA probes achieved in different stages of synthesis, including alkyne modified DNA, DNA-1-Azidoadamantane (DNA-AA), and DNA-AA@cucurbit[6]uril hydrate (DNA-AA@CB[6]), were first analyzed for their translocation behaviors in α-hemolysin (α-HL) nanopores, respectively. DNA-AA molecules were obtained through a click reaction of 1-Azidoadamantane (AA) to alkyne modified DNAs in the presence of Cu ions, and DNA-AA@CB[6] complexes were further obtained by a host-guest reaction between DNA-AA and CB[6]. Prior to analyzing the DNA probes, the conductivity of an open α-HL nanopore in the lipid bilayer membrane was probed using a single channel conductance setup (Fig. 2a). Upon insertion of a single α-HL nanopore into the membrane from the *cis* side, an instant current increase can be observed as shown in the current trace under either a positive or a negative voltage bias (Supplementary Fig. 2a). Notably, the insertion orientation of the α-HL in the lipid bilayer is pivotal for observing signature current events of DNA probes[40]. A properly inserted α-HL pore is characterized by larger ionic current under a positive *trans* voltage than it is under a negative *trans* voltage, i.e., $279.4 \pm 5.0$ pA under +100 mV, and $-241 \pm 6.6$ pA under -100 mV[19,41]. The suitable thickness of a bilayer membrane for insertion is ~5 nm, corresponding to 160-180 pF capacitance (Fig. 2b, calculation details shown in Supplementary Information). The electrolytic conductance was examined under a ramping voltage to ensure that the conductivity can afford efficient translocations of DNA probes through the α-HL nanopore to induce a large number of blockades to the ionic current (Supplementary Fig. 2b). When the applied *trans* voltage is above 160 mV, signals of each subpopulation of the DNA probe synthesis (alkyne modified DNA, DNA-AA, and DNA-AA@CB[6]) can be efficiently observed in the raw current traces (Fig. 2c). Representative current traces of alkyne modified DNA, DNA-AA, and DNA-AA@CB[6] (Fig. 2d, Supplementary Fig. 3) clearly show differences between these signals induced by the structural evolution of the DNA probes. In comparison with alkyne modified DNA and DNA-AA, the DNA-AA@CB[6] signal has an unique pattern that is characterized by a long deep blockage (Level 1) induced by the translocation of the DNA backbone, and a short oscillation (Level 2) induced by the dissociation of the CB[6] inside the nanopore (Fig. 2e). This pattern is distinctly identifiable from signals of other molecules in this system, which is utilized in this study to ensure highly specific detection. The sensitivity of the detection is afforded by the conversion of p24 antigen to the probe through a DNA click reaction. To quantitatively characterize the evolution of DNA probes, current blockage events of products at each step of the click reaction were statistically analyzed for their current blockage ($I/I_O$) and dwell time ($\tau$). As shown in Fig. 2f, alkyne modified DNA causes distinct

blockage events with blockades centered at $0.855 \pm 0.001$. Observed nonspecific spikes ($I/I_O$ at $0.723 \pm 0.008$ and $0.192 \pm 0.002$) can be attributed to occasional random DNA collision with the nanopore or background noise. After the click reaction, the product DNA-AA was found to have a blockade profile ($0.804 \pm 0.003$) similar to that of the alkyne modified DNA, with significantly reduced number of non-specific spikes. However, its mean dwell time was significantly increased from $1.94 \pm 0.50$ ms to $19.34 \pm 0.89$ ms (Fig. 2g, $p < 0.0001$, $\tau_{DNA}$ vs. $\tau_{DNA-AA}$ by two-tailed unpaired Student t-test). A slight overlap between the contour profiles of alkyne modified DNA and DNA-AA (Supplementary Fig. 4) could be attributed to the false signals caused by the returning of trapped analytes from the nanopore vestibule to the *cis* solution without translocation[42]. Although the introduction of CB[6] to DNA-AA was not able to further change its current blockade and dwell time profile, the DNA-AA@CB[6] structure causes a two-stage translocation as previously discussed, that results in a unique signal with two levels of current blockade at $0.948 \pm 0.027$ and $0.723 \pm 0.037$ (Fig. 2h) to afford specific and reliable recognition of probe signals from other signals. Excessive free CB[6] in the solution cannot generate multi-level signature signals, thus will not affect its specificity to the DNA-AA@CB[6] probe (Supplementary Fig. 5).

In the capture rate (i.e. translocation frequency) study, standard DNA-AA@CB[6] probe samples were tested using independent nanopores. Events induced by DNA-AA@CB[6] were extracted by observing their multi-level and oscillation characteristics and counted. Cumulative counting of the multi-level signature events shows the same increase rate of event numbers over different recording times (1, 2, 3, 5 min) and across four different pores for the same sample (Fig. 2i), which allows capture rate calculation by dividing the multi-level event number over the recording time. While the total single-molecule event rate (alkyne modified DNA, DNA-AA, and DNA-AA@CB[6]) remains consistent at >500 min⁻¹, positive correlation was observed between the capture rate of multi-level events and the Cu ion concentration used to form DNA-AA. The Cu ion concentrations tested (0.1, 0.5, 1 mM) were anticipated to cover most clinical samples (Fig. 2j). The DNA-AA@CB[6] capture rate is also closely related to the applied voltage: less than 2 min⁻¹ multi-level events was observed with 120 mV or lower applied voltage, but it rapidly increased to a maximum of ~40 min⁻¹ when the applied voltage was ramped up to 160 mV or above (Fig. 2k and Supplementary Fig. 6). While no multi-level signature event was detected with an increasing negative voltage (Supplementary Fig. 7), these results indicate that a higher positive voltage can promote the dissociation of the DNA-AA@CB[6] complex and reduce the possibility of DNA escaping back to the *cis* side. However, high voltage can also weaken the stability of the lipid bilayer membrane, which is shown by the increasing standard deviation and current leakage (Fig. 2k and Supplementary Fig. 8). In this study, 160 mV operating voltage and 2-minute recording time was employed to optimize the sample-to-answer time while maintaining membrane stability as well as consistent capacitance (160-180 pF) and interpore capture rate (Fig. 2l).

The capture rate of the DNA-AA@CB[6] probes also strongly depends on the electrochemical environment of the nanopore system, which plays a key role in improving the detection sensitivity. To this end, we also studied the effects of various electrolyte concentrations and pH of the working solution to the frequency of multi-level signature events[41]. Although concentration gradient was found to increase translocation of linear macromolecules[28,43], our results show that optimal multi-level signal frequency can only be achieved using balanced *cis* and *trans* work solutions (i.e. with the same electrolyte concentration and pH), and that any disruption of the balance could result in decreased signal frequency (Supplementary Fig. 9). Therefore, 3 M KCl work solution (10 mM Tris, pH 8.0) in both *cis* and *trans* sides were employed for all following experiments.

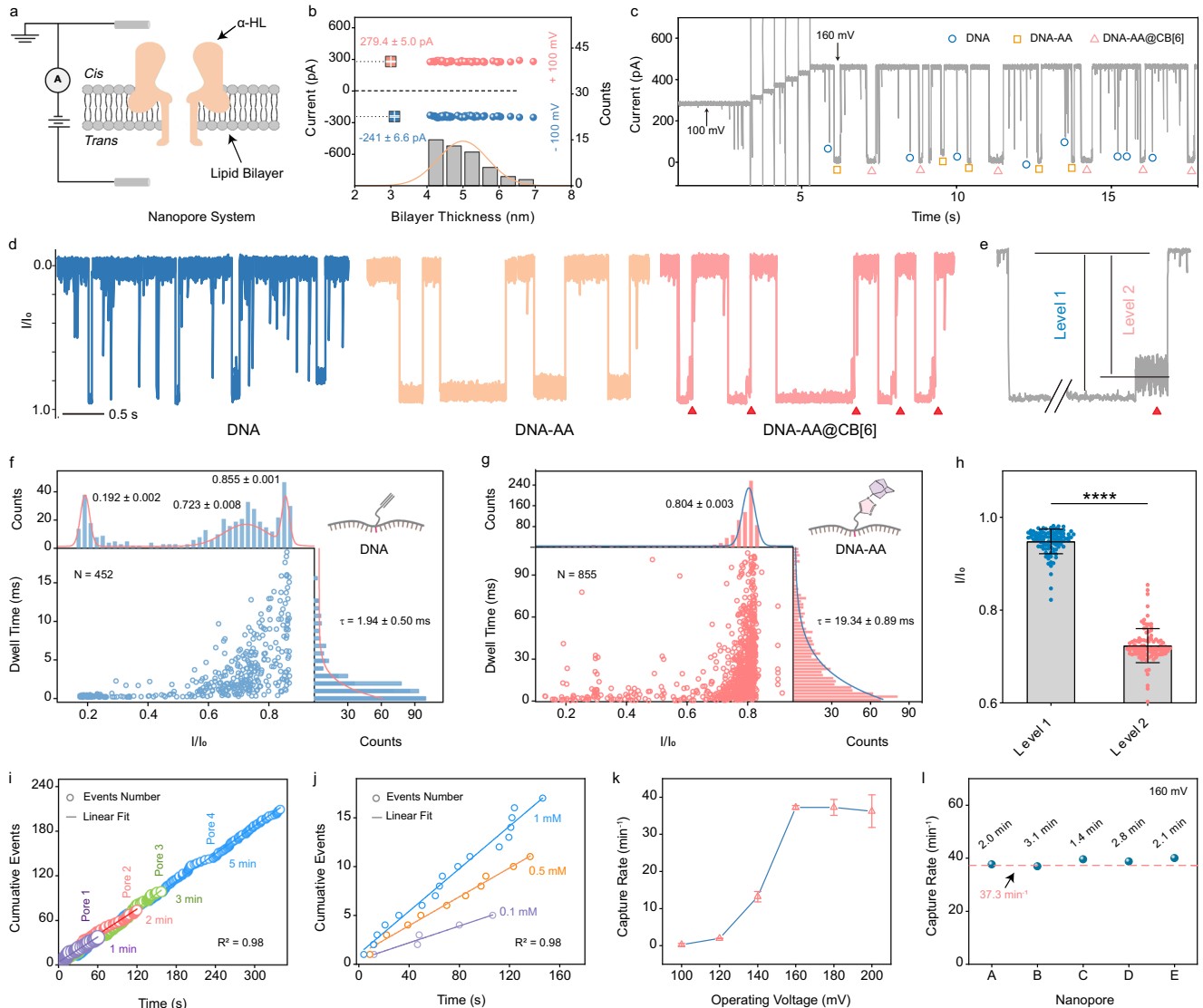

**Fig. 2 | Characterization of nanopore and multi-level signature events.**
**a** Schematic illustration of an α-HL protein inserted into a lipid bilayer membrane with an applied external voltage. The potential across the bilayer is applied to the *trans* side through Ag/AgCl electrodes, while the *cis* side is grounded.
**b** Corresponding average open pore current values at ±100 mV and the distribution of calculated thicknesses of lipid bilayer membranes ($n = 50$ biologically independent experiments). **c** Continuous raw current traces of DNA probes achieved in different stages of synthesis. Circles, squares, and triangles indicate alkyne modified DNA, DNA-AA, and DNA-AA@CB[6], respectively. **d** Representative current traces of alkyne modified DNA, DNA-AA, and DNA-AA@CB[6] translocations, respectively. **e** A typical multi-level signature signal generated by translocation of a DNA-AA@CB[6] probe. Each signal event is characterized by a pattern with Level 1&2. **f**, **g** Two-dimensional scatter plots of **f** alkyne modified DNA and **g** DNA-AA translocation signals, respectively. Top: histograms of $I/I_O$ with Gaussian fitting.

Right: histograms of dwell time with exponential fitting. **h** Average current blockades of Level 1&2 in multi-level current events. ****$p < 0.0001$ (Level 1 *vs.* Level 2 by two-tailed unpaired Student $t$-test, each point represents the blockade of Level 1 or 2 for a multi-level current event, $n = 100$ signals from 20 independent pores).
**i**, **j** Cumulative numbers of multi-level events acquired using different independent nanopores **i** over different recording time and **j** with different Cu ion concentrations (total events $N > 500$ min⁻¹). **k** Capture rates of DNA-AA@CB[6] multi-level events under increasing *trans* potentials from 100 mV to 200 mV. Data represents mean ± SD of replicates ($n = 3$ biologically independent experiments, total events $N > 1000$ for each data point). **l** Capture rates of multi-level events recorded using five independent nanopores. Data was acquired using 3 M KCl, 10 mM Tris buffer at pH 8.0 and under 160 mV *trans* potential unless otherwise specified. Source data are provided as a Source Data file.

## Cu⁺ Ion Concentration Dependent Catalytic Click Reaction

Further optimization was focused on the click reaction between azide and alkyne, which is an essential step in the formation of DNA-AA@CB[6] probes[44]. This reaction can be effectively catalyzed under ambient conditions by Cu⁺ ions (Fig. 3a), which were obtained by sodium ascorbate reduction of Cu²⁺ ions released from CuONPs[45,46]. Mass spectrometry (MS) confirms the transformation of alkyne modified DNA to DNA-AA after the click reaction (Fig. 3b). The reaction yield is quantitatively related to Cu⁺ ion concentration with virtually no by-products[47]. No DNA-AA signal can be detected in absence of Cu⁺ ions with extended reaction time up to 12 h (Fig. 3c). The concentration

study indicates increasing DNA-AA signals when Cu⁺ ions were increased from 0 to 10 mM (Fig. 3d). Further kinetic study shows that the yield in this reaction system is also positively correlated to reaction time under various Cu⁺ ion concentrations (Fig. 3e and Supplementary Fig. 10). In nanopore measurements of DNA-AA@CB[6] formed with excessive CB[6], an optimal incubation time of 4 h was concluded for maximizing multi-level signal frequency (Supplementary Fig. 11a). To eliminate the reaction time effect and achieve optimal catalyst-product linearity, click reaction time was limited to 4 h by EDTA termination in all following studies. Although the reaction efficiency plateaued at 10 mM of Cu⁺ ions, a good linear relationship was

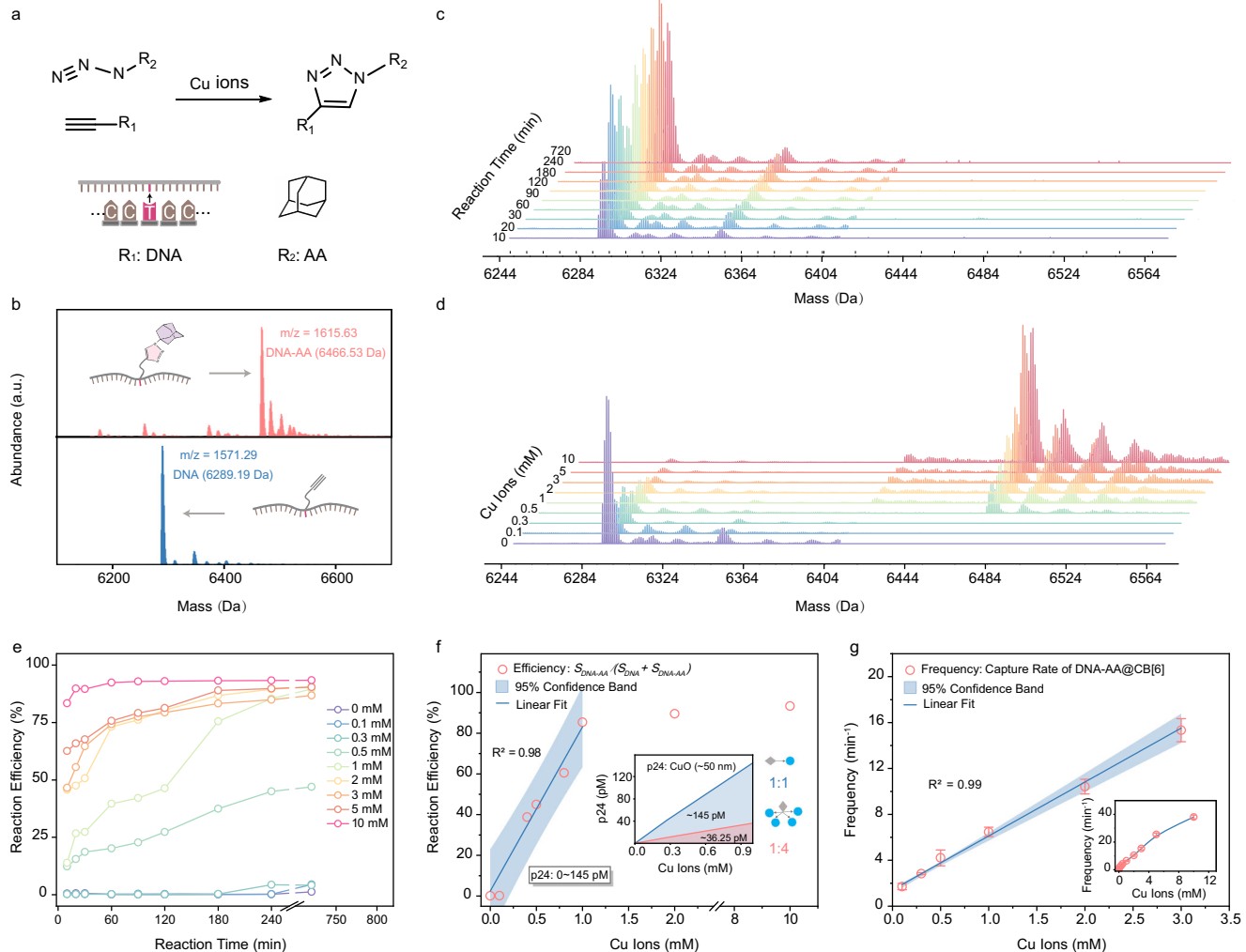

**Fig. 3 | Click reaction catalyzed by Cu⁺ ion. a** Scheme of Cu⁺ ion-catalyzed click reaction between alkynes ($R_1$) and azides ($R_2$). **b** MS characterizations of the substrate (alkyne modified DNA) and the product (DNA-AA) of the click reaction. **c**, **d** MS characterizations of the substrate and the product of the click reaction **c** under different reaction time in absence of Cu⁺ ions and **d** under the same reaction time with different Cu⁺ ion concentrations. **e** Reaction efficiency as a function of reaction time in presence of various Cu⁺ ion concentrations. Reaction efficiency is determined by $S_{DNA\text{-}AA}/(S_{DNA} + S_{DNA\text{-}AA})$ (S: integrated peak area in mass spectrum). **f** Reaction efficiency as a function of Cu⁺ ion concentrations under 4 h reaction time with linear fit between 0 mM and 1 mM Cu⁺ ions. Each data point

represents an independent experiment. Solid line indicates linear regression ($R^2 = 0.98$). Shadow indicates the 95% confidence interval of the fitted line. Inset shows the theoretical p24 antigen concentration range covered. **g** Correlation between multi-level signal frequency and Cu⁺ ion concentration. Inset shows the correlation in higher range up to 10 mM. Data represents mean ± SD of replicates ($n = 3$ biologically independent experiments, total events $N > 1000$ for each data point). Solid line indicates linear regression ($R^2 = 0.99$). Shadow indicates limits of 95% confidence interval of the fitted line. Source data are provided as a Source Data file.

obtained between 0 mM and 1 mM Cu⁺ ions (Fig. 3f). Theoretical p24 concentrations corresponding to this range are 0-145 pM and 0-36.25 pM for 1:1 and 1:4 p24 antigen-CuONP ratios, respectively (Inset of Fig. 3f), to cover quantification needs in most clinical scenarios (0.42-33.3 pM)[48–50]. While the Cu⁺ concentration dependency of DNA-AA formation was clearly characterized using HPLC (Supplementary Fig. 12a) and MS, nanopore measurements, on the other hand, cannot completely separate subpopulations of alkyne modified DNA and DNA-AA translocation events due to signal similarity (Supplementary Fig. 12b). Upon introduction of CB[6] to the click reaction product, the host-guest interaction between CB[6] and DNA-AA can be deduced *via* nuclear magnetic resonance (NMR) results showing the chemical shift of protons of CB[6] molecules (Supplementary Fig. 11b). Under these optimized reaction conditions, DNA-AA@CB[6] products formed under various Cu⁺ ion concentrations were measured by nanopore. Muti-level signal frequencies showed an excellent linear correlation ($R^2 = 0.99$) with Cu⁺ ion concentrations used to fabricate DNA-AA@CB[6] probes (Fig. 3g and Supplementary 12c).

## Optimization of the CAN Assay

We further exploited the use of the biotin-streptavidin linker in the assay to increase the amount of CuONPs binding to each p24 antigen from 1 to 4 in order to lower the LOD (Fig. 4a)[51]. Two sandwich structures with (route II) and without (route I) the biotin-streptavidin linkers were formed in p24 spiked human sera. In route II, CuONPs were linked to detection antibodies using a streptavidin-biotin-polyethylene glycol (PEG) complex through copper-thiol adsorption chemistry[52–54]. The biotin-PEG linker was synthesized in-house with commercially available reagents through addition reaction between the double bond in maleimide and the sulfhydryl (Supplementary Fig. 13). After magnetic separation and acid treatment, Cu ion quantification by inductively coupled plasma-optical emission spectroscopy (ICP-OES) indicated that, with similar amount of Fe ions, the sample from route II contains ~4× higher amount of Cu ions than the route I sample (Fig. 4b). Furthermore, transmission electron microscopy (TEM) and dynamic light scattering (DLS) results confirmed stable aggregation of CuONPs through biotin-streptavidin linkage in route II (Fig. 4c, d). The synthesis

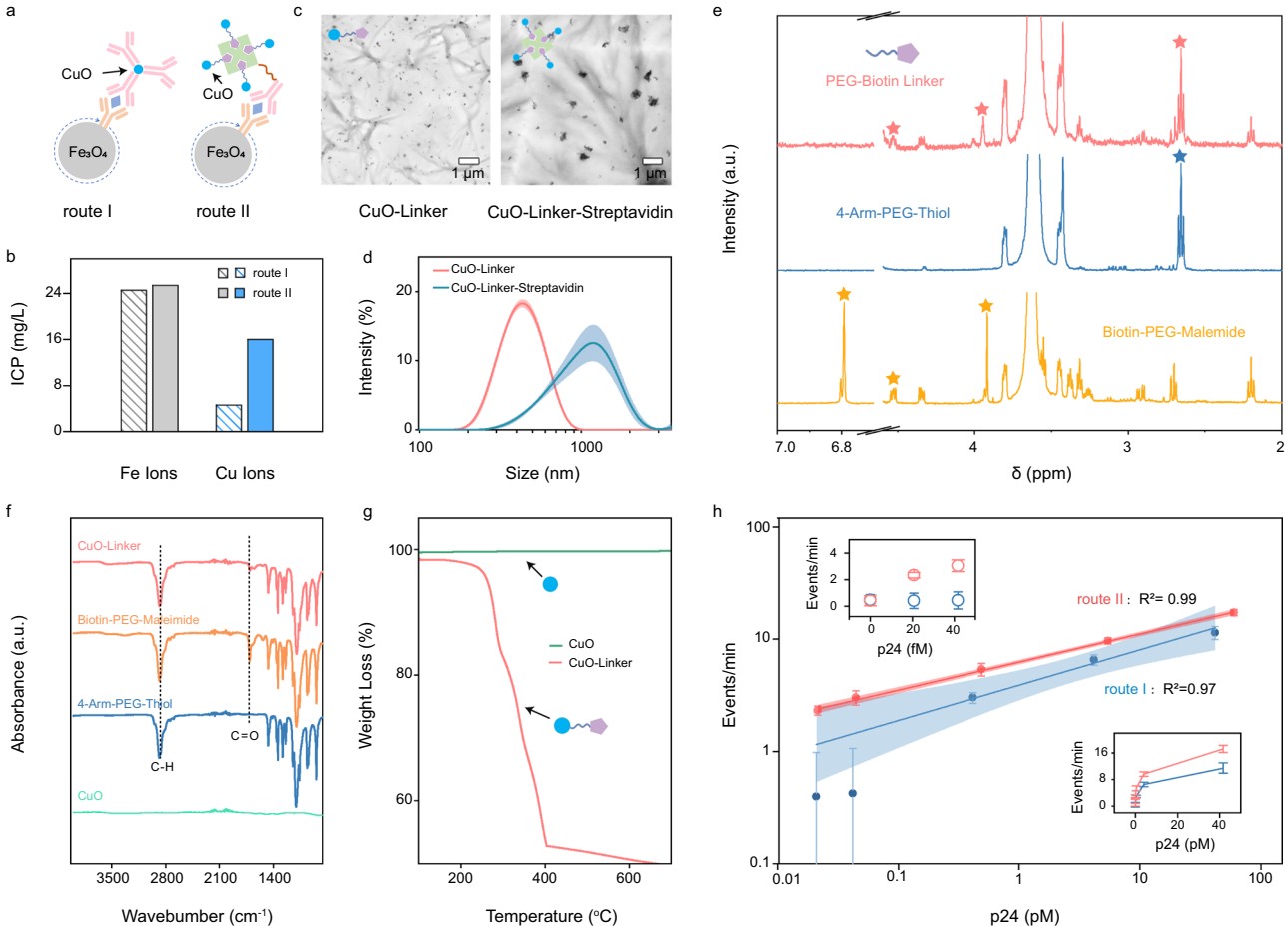

**Fig. 4 | Assay optimization and analytical performance. a** Schematic illustration of two sandwich structures obtained for assays with the traditional route (I): MBs-p24-CuO sandwich structures and the amplified route (II): MBs-p24-CuO sandwich structures with biotin-avidin amplification. **b** Corresponding ICP-OES results of Cu and Fe ions of both sandwich structures after acid treatment. **c** Representative TEM images (*n* = 3 independent replicates were conducted with similar results) of CuONP-Linker and CuONP-Linker-Streptavidin structures. **d** Hydrodynamic size distribution profiles of CuONP-Linker and CuONP-Linker-Streptavidin structures. Data represents mean ± SD of replicates (*n* = 3 biologically independent samples. Shadows indicate error bands. **e** ¹H NMR (400 MHz) spectra of reactants (middle and lower) and achieved PEG linkers (upper). **f** FTIR spectra of bare CuONPs, Linker, and CuONP-Linker structures. **g** thermogravimetric curves of bare CuONPs and

CuONP-Linker structures. **h** Correlations between multi-level signal frequency and p24 concentration in human serum within the range of 0-46.7 pM (0−1000 pg/mL, final concentration: 0, 0.5, 1, 10, 100, 1000 pg/mL) processed using route I and route II assays, respectively. Insets show the linear correlations within lower range (0-50 fM) and wide range (0-40 pM). Data represents mean ± SD of replicates (*n* = 5 biologically independent experiments, total events *N* > 2000 for each data point, detailed information shown in Supplementary Table 1). Solid line indicates linear regression with slopes of 0.315 ± 0.045 (route I, R² = 0.97, equation: y = 0.590x^{0.315}) and 0.261 ± 0.004 (route II, R² = 0.99, equation: y = 0.818x^{0.261}). Shadow indicates limits of 95% confidence interval of the fitted line. Source data are provided as a Source Data file.

of PEG linkers for linking CuONPs and biotins was characterized by NMR. According to ¹H NMR spectra (Fig. 4e), disappearance of the olefinic signal of the maleimide moiety in at δ6.7 ppm strongly corroborated the coupling reaction between Biotin-PEG-Maleimide and 4-Arm-PEG-Thiol, and the presence of biotin signals (δ3.9 and δ4.5 ppm) and residual thiol signal (δ2.6 ppm) confirmed the production of the final PEG-Biotin linkers, which will enable the conjugation of CuONPs and streptavidinized antibody[55,56]. Fourier-transform infrared (FTIR) spectra with two absorption bands at 2866 cm⁻¹ and 1700 cm⁻¹ attributed to stretching bonds of C-H and C = O in PEG and biotin on CuONP surface (Fig. 4f)[57], together with a weight loss of approximately 52% at 400 °C (Fig. 4g), confirmed the immobilization of PEG-Biotin linkers onto the CuONP surface.

Next, CAN assays constructed by both synthesis routes were used to establish calibration curves using healthy donors' sera spiked with various amount of recombinant p24 antigens (Fig. 4h). A positive correlation between multi-level event frequencies and p24 concentrations were observed in both assays (inset of Fig. 4h, wide range).

However, in comparison with route I in which accuracy and reproducibility were greatly affected by low signal frequency, the biotin-streptavidin amplification strategy (route II) greatly improved detection accuracy and reproducibility, especially in the low p24 concentration range (0-42 fM), to achieve approximately 20-fold lower LOD than that of route I assay (inset of Fig. 4h, lower range) while reliably quantify changes in p24 concentration. We used the Evaluation of Detection Capability for Clinical Laboratory Measurement Procedures (EP17 protocol) to further verify the LOD of the route II assay, which was determined using the limit of blank (LOB) and standard deviation (SD) of replicate tests of a sample known to contain a low concentration of analyte. Calculation of the LOB and the LOD follows previously established equations: $LOB = mean_{blank} + 1.645(SD_{blank})$, $LOD = LOB + 1.645(SD_{low\ concentration\ sample})$[58]. The calculated LOD (0.44 pg/mL, 18.3 fM) agrees with our calibration curve. An inspection of the raw current trace data recording also confirmed that multi-level signature events can be reliably detected in the 0.5 pg/mL (20.8 fM) but not in the blank control sample (Supplementary Fig. 14).

## Analytical Performance Benchmarking

Recently, various approaches ranging from NP-based HIV biosensors to advanced ELISAs have been developed to detect p24 in human serum, but most of them are yet under diverse clinical studies[16,59–61]. For comparison, a newly developed fluorescent nanocluster-based method and a traditional ELISA were performed side-by-side with our CAN assay. In the fluorescent nanocluster-based method, two types of glutathione (GSH) functionalized CuNCs with blue and red fluorescence were synthesized, characterized, and used to detect p24, respectively (Supplementary Figs. 1, 15a, b)[16,62]. Their simple chemical compositions allow easy engineering and optimization[16,63]. By conjugating fluorescent CuNCs to the end of an antibody-antigen-antibody complex, p24 antigens can be detected using the fluorescence from CuNCs (Supplementary Fig. 16a, d). Photoluminescence spectra and excitation spectra of these two types of GSH stabilized CuNCs indicate excitation at 370/360 nm and emission at 440/630 nm in aqueous solution, respectively (Supplementary Fig. 16b, e). Positive correlation between the photoluminescence intensity and the concentration of CuNCs suggest that p24 antigens can be quantified by measuring the fluorescence intensity (Supplementary Fig. 15c). Upon establishing calibration using p24 spiked sera, analytical LODs for the blue and the red fluorescent CuNCs assays were found to be 50 pg/mL (2.1 pM) and 20 pg/mL (0.83 pM), respectively (Supplementary Fig. 16c, f). Higher analytical sensitivity of the red fluorescent assay can be attributed to lower interference of background fluorescence[64]. In another paralleled control experiment, an ELISA based fourth generation HIV-1 testing kit was able to reliably detect p24 at 10 pg/mL (0.41 pM), demonstrating 2-5-fold higher analytical sensitivity than CuNCs based assays (Supplementary Fig. 16g, h). Additional comparisons of LOD, dynamic range, and clinical validation with more techniques in previous reports were also summarizes to further highlight the superiority of the CAN assay (Supplementary Table 2 and Fig. 17).

## CAN Assay Clinical Validation in a Pilot Cohort

Among all methods experimented, the optimized CAN assay exhibited superior analytical sensitivity. Its clinical performance was further evaluated using 124 human samples (118 eligible) from a pilot HIV cohort. The ELISA based fourth generation HIV-1 test was also evaluated as a reference. Flow diagram describes the disposition of the study subjects (Supplementary Fig. 18). Based on viral load results obtained at the same time of the sera collection, subjects were divided into four groups: Negative group (clinical state: Non-HIV), VL-0 group (HIV/AIDS stage with undetectable viral load), LOD-30 group (HIV/AIDS stage with viral load between LOD and 30 copies/mL), and Positive group (HIV/AIDS stage with >30 copies/mL viral load). Each sample was tested by the CAN assay and a commercialized fourth generation HIV-1 ELISA testing kit (Fig. 5a and Supplementary Fig. 19). While CAN results in general agree to ELISA results among subjects in the Negative group, the CAN assay showed superior sensitivity that allows p24 detection in patients missed by nucleic acid and/or ELISA assays in VL-0 and LOD-30 groups. Particularly, in clinically diagnosed patients with detectable viral load (LOD-30 and Positive groups), the CAN assay was able to detect p24 in 48 of 55 samples (87.3%) and 19 of 19 samples (100%), respectively, while the sensitivity of ELISA for these two groups is only 18.2% and 42.1% (Fig. 5b, c). The sensitivity improvement and the quantification capacity of the CAN assay also enables potential prognostic uses. Statistical analysis among three groups (Negative, VL-0 group, and LOD-30) shows that p24 was at similarly low level in Negative and VL-0 groups, but was significantly higher in patients with detectable (LOD-30) viral load. Meanwhile, no significant difference was detected between p24 across these groups measured by ELISA. ELISA was only able to detect significantly higher p24 as CAN did in patients with high (positive) viral load when they were added to the analysis (Fig. 5d, e). To further compare the sensitivity and the specificity of the CAN assay and ELISA, receiver-operating characteristic (ROC) curves were constructed (Fig. 5f) based on results from these two assays for diagnosing of HIV and AIDS. In each ROC curve, a cut-off value was determined to maximize the sensitivity and the specificity comprehensively[65]. The area under the curve (AUC) of CAN-HIV and CAN-AIDS curves were significantly higher than those of ELISA curves, i.e., $0.95 \pm 0.02$ (CAN-HIV, $p < 0.0001$) and $0.90 \pm 0.04$ (CAN-AIDS, $p < 0.0001$) comparing to $0.63 \pm 0.07$ (ELISA-HIV, $p = 0.16$) and $0.66 \pm 0.08$ (ELISA-AIDS, $P = 0.10$), respectively. Using 2.29 fM and 17.01 fM as cut-off values, the CAN assay achieved 89.47% and 80.65% detection sensitivity in HIV and AIDS cohorts, respectively, with 100% specificity. CD4 cell count/percentage is an important immunological marker that indicates stage of infection and informs antiretroviral therapy. In patients with a valid CD4 count/percentage, correlation observed between CAN measured p24 levels and CD4 counts/percentage agrees with previous reports (Fig. 5g and Supplementary Fig. 20)[66,67]. Among patients categorized by CD4 counts, average p24 level was significantly higher in the severe immunosuppression group ($\leq 200/mm^3$ CD4) than in the normal CD4 count group (CD4 $\geq 500/mm^3$) (Supplementary Fig. 21). In patients with quantifiable viral loads, p24 also positively correlated with viral load (Fig. 5h), in consistency with previous reports[3,49].

## Discussion

The viral capsid protein p24 is recognized as an alternative virological biomarker for HIV infection. Quantification of p24 at ultra-low concentrations is not only of great clinical significance for early detection and prognostic monitoring of HIV patients, but also expected to elucidate the relationship between p24 antigen and HIV viral load during acute infection. However, clinically employed ELISA kits are suboptimal for these purposes due to their inadequate detection sensitivity. Over the past decades, many biosensors have been developed using NPs, as they offer simple and efficient surface conjugation for target enrichment and sensing. Most recently, Cu fluorescent NCs have attracted extensive research attention not only because of their high yield in mild synthetic conditions, but also due to the abundance and low cost of Cu, which offers potential for large-scale production. Although most NC based biosensors demonstrated high sensitivity and excellent stability, there remain many challenges ahead: (1) fluorescent NCs with high quantum yield and surface suitable for biological modification are needed; (2) near-infrared luminescent NCs to avoid interference from background fluorescence remains to be developed;[68] (3) separation of signal detection from the assay process in order to reduce the influence of the assay process on the signal reading, as well as confining infectious pathogens in the assay process for safety.

To this end, we have developed a click chemistry amplified nanopore (CAN) assay and validated its performance for quantifying HIV-1 p24 antigen in clinical samples. The design of this assay can be attributed to the following inspirations: (1) Nanopore has outstanding sensitivity towards nucleic acids; (2) The unique multi-level signal induced by the probe DNA endows excellent specificity; (3) Conversion of p24 to probe DNAs with amplifications through a sandwich assay by a $Cu^+$ ion catalyzed azide-alkyne click reaction. Based on these revelations, the MBs-p24-CuO sandwich structures were engineered and constructed for the nanopore assay. A streptavidin-biotin reaction was employed to further increase the CuO loading per p24 antigen, while simplifying the assay protocol. PEG-linkers containing multiple anchoring groups was used to conjugate biotins to CuONPs for increased stability in biological environments due to resistance to linker displacement by free thiols[69,70]. Upon acid treatment of the sandwich structures, released $Cu^+$ ions catalyze the production of numerous DNA-AAs through a click reaction, followed by an incubation with CB[6] to form the DNA-AA@CB[6] probes. After revealing relationships between p24, $Cu^+$ ions, and host-guest DNA probes in the assay process, the calibration curve established by frequencies of

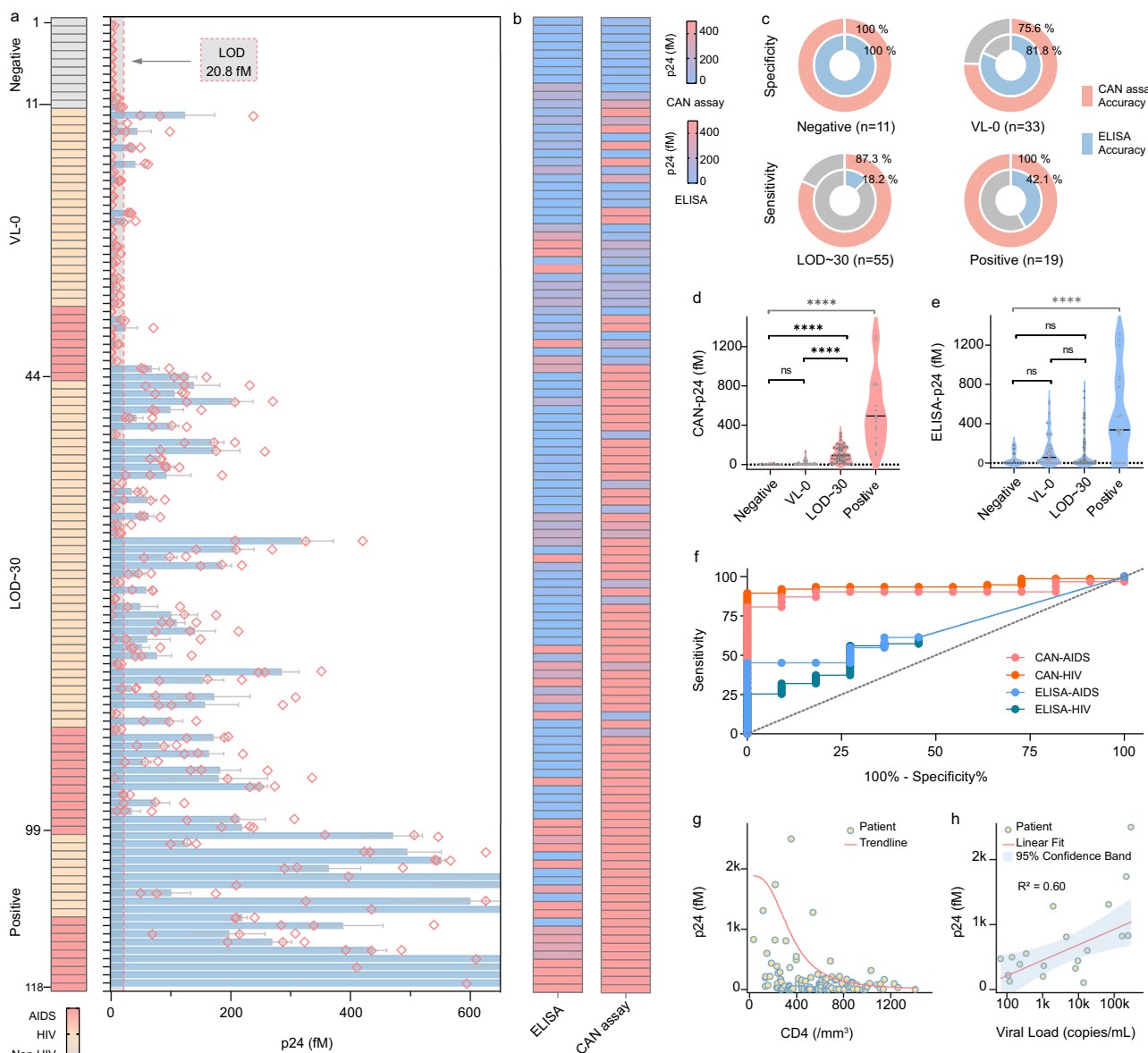

**Fig. 5 | Clinical validation in a pilot cohort. a** Quantitative measurements of p24 in clinical samples using the CAN assay, with the clinical diagnosis of each patient. Bar graph data represents mean ± SD of replicates (*n* = 3 biologically independent experiments, total events *N* > 1000 for each sample, detailed information shown in Supplementary Table 3). Each red diamond represents the result from one experiment. The dashed line indicates the LOD of the CAN assay. **b** Cluster map produced from p24 results measured by the CAN assay and ELISA for 118 eligible patients. Blue color indicates results below LOD. **c** p24 detection accuracy for the CAN assay and ELISA in different patient groups using VL as reference (Negative: *n* = 11; VL-0: *n* = 33; LOD-30: *n* = 55; Positive: *n* = 19). **d, e** Average levels of p24 concentrations obtained by **d** the CAN assay and **e** ELISA for different patient groups. Each point represents an independent clinical individual (Negative: *n* = 11; VL-0: *n* = 33; LOD-30: *n* = 55; Positive: *n* = 19). Each solid line represents mean p24

value of all individuals in that group. *p* values were calculated by one-way ANOVA with post-hoc Tukey tests among 3 (black) or 4 (gray) groups. **** indicates *p* < 0.0001, ns indicates no significant difference. For CAN assay, *p* (Negative *vs.* VL-0) = 0.76, F (11,33) = 17.3; *p* (Negative *vs.* LOD-30) <0.0001, F (11,55) = 101.0; *p* (VL-0 *vs.* LOD-30) <0.0001, F (33,55) = 101.0; *p* (Negative *vs.* Positive) <0.0001, F (11,19) = 709.2. For ELISA, *p* (Negative vs. VL-0) = 0.32, F (11,33) = 126.1; *p* (Negative *vs.* LOD-30) = 0.44, F (11,55) = 109.0; *p* (VL-0 *vs.* LOD-30) = 0.89, F (33,55) = 109.0; *p* (Negative vs. Positive) <0.0001, F (11,19) = 481.7. **f** ROC curves for p24 detection by the CAN assay and ELISA in HIV and AIDS populations. **g, h** Correlation between **g** CD4 counts or **h** viral loads and CAN measured p24 concentrations. Each point represents an independent clinical individual. Solid line indicates **g** nonlinear logistic fit or **h** linear regression (R² = 0.60). Shadow indicates limits of 95% confidence interval of the fitted line. Source data are provided as a Source Data file.

multi-level signals against p24 concentrations in human serum exhibited a wide dynamic range (0.5-1000 pg/mL, 0.02-46.7 pM), a reliable precision to quantify p24 concentration, and a 20-100-fold lower LOD when compared to fluorescent CuNCs-based assays and a commercialized ELISA. In clinical validation, the CAN assay was applied to quantify p24 antigens in samples of 118 patients from various groups. Further analysis of clinical results not only showed superior sensitivity of the CAN assay in samples missed by traditional viral load and/or

ELISA assays, but also demonstrated its reliability in correlation with CD4 count and viral load results, which warrants future investigations of the CAN p24 antigen assay in large-scale cohorts for HIV-1 early detection and prognosis.

For translational applications, the current strategy still needs several improvements. First, the assay sensitivity may be further improved by employing a higher transmembrane voltage during measurements to increase the probe DNA capture rate, prevent probe

escaping through the nanopore entrance, and promote the dissociation of the host-guest structure (DNA-AA@CB[6]) in the pore. However, this is limited by the fragile lipid membrane that supports the nanopore protein. To circumvent this issue, more robust liposome membranes, polymer bilayers, or solid-state membranes that can maintain integrity under high voltages are highly desirable[71–76]. One promising alternative is to replace the lipid membrane with a copolymer membrane[77], or make a hybrid system by nesting an α-HL protein into a solid-state nanopore in order to achieve robustness and device compatibility, while maintaining the high analytical performance[78]. Secondly, elevated standard deviations were observed in clinical results as normally seen in previous clinical diagnostics studies[79,80], especially in the lower p24 range, which may cause false calling of results close to LOD. This is likely due to the relatively low event counts in these samples and the higher complexity of interferents in HIV patients' sera comparing to standard spiked sera samples. Strategies for improving probe DNA capture rate such as increasing the driving force using higher voltage and probes with higher charges, increasing the p24 to probe DNA ratio using advanced particle materials and linkers, as well as reducing the sample complexity by separating common blood interferents are currently under study. In addition, the current assay protocol still involves multiple hands-on steps, leading to long turn-around and potential human errors. Assay automation by microfluidic technologies needs to be incorporated to reduce both time and financial costs of this method, especially for future point-of-care applications[81]. By addressing these issues, we believe that the CAN assay can afford an ultrasensitive, rapid and robust method for HIV-1 early diagnosis and prognosis, and can also be applied to mitigate any future infectious disease outbreaks with minimum modification.

## Methods

### Ethical statement
This study was approved by the University of South Carolina Institutional Review Board for Human Research (Approval No. Pro00083810). All research was performed in accordance with relevant guidelines and regulations. All participants have provided informed consent. Participants were not compensated. Study results were not used in any ways in clinical procedures and were not made available to participants.

### Route I - Conjugation of detection antibodies on CuO nanoparticles
CuONPs (1 mg) were dispersed in 1 mL PBS by ultrasonication for 10 min and then incubated with detection antibodies (100 μg, 1.31 mg/mL, undiluted anti-HIV-1 p24 antibody [38/8.7.47], Abcam, Cat#: ab9044) for 1 h with vortex (30 x $g$). After centrifugation at 3500 x $g$ for 5 min to remove the supernatant, antibody functionalized CuONPs were washed twice and redispersed in 1 mL PBS solution, then blocked with 200 μL BSA (10% in PBS) for 30 min, and finally stored at 4 °C for further use[82].

### Route I - Immunoprecipitation of p24 antigens from human serum and conversion to Cu ions
Capture antibody-coated MBs (20 μL in PBS, Supplementary Methods) were washed with 500 μL assay buffer for three times and dispersed in 500 μL diluted healthy donor serum (human serum: assay buffer = 1:1, v/v). To establish a calibration curve, standard samples were made by adding various amount of p24 antigens to diluted sera and vortexing for 30 min at room temperature to reach final concentrations of 0, 0.5, 1, 10, 100, 1000 pg/mL. Next, antibody-modified CuONPs (200 μL) were mixed with each standard sample and vortexed for another 30 min at room temperature. After forming sandwich structures, MBs were magnetically separated and washed three times with washing buffer. To release Cu ions, sandwich structures were then treated with 200 μL HCl (0.1 M) for 10 min with vortex, and washed with 200 μL

washing buffer for 5 times. All supernatants were collected and concentrated to 40 μL for constructing probe DNAs.

### Route II - Conjugation of detection antibodies to streptavidin
The conjugation experiment follows a simple and rapid procedure provided along with the Streptavidin Conjugation Kit (Lightning-Link) which targets primary amine groups (e.g. lysines). Briefly, streptavidin modifiers and detection antibodies (100 μg, 1.31 mg/mL, undiluted anti-HIV-1 p24 antibody [38/8.7.47], Abcam, Cat#: ab9044) were mixed and incubated for 3 h at room temperature without light. The result solution was then quenched for 30 min and stored for future use without purification.

### Route II - Synthesis of trithiol-PEG-biotin linkers
Biotin-PEG was prepared following a previously reported method with slight modifications[83]. Briefly, 200 μM 4-arm-PEG-thiol solution (Laysan Bio, MW: 10,000 Da) was added to 200 μM biotin-PEG-maleimide solution (Laysan Bio, MW: 5000 Da). The mixture was incubated on a shaker for 24 h at room temperature, followed by filtration using a 10,000 Da cutoff filter.

### Route II - Conjugation of biotin-PEG on CuO nanoparticles
CuONPs (1 mg) were dispersed in 1 mL PBS, and then incubated with 10 mL biotin-PEG linker solution on a shaker for 24 h at room temperature. After the incubation, assay buffer was added to bring the final NaCl concentration to 0.05 M. After an hour, the NaCl concentration was raised to 0.3 M using 5 M NaCl, and the mixture was incubated with shaking at room temperature for an additional hour. The resulted mixture was aliquoted into low retention tubes and centrifuged at 3500 x $g$ for 10 min at room temperature. After removal of the supernatant and two additional washes with DI water, biotin-PEG-CuONPs were resuspended in 1 mL assay buffer and stored at 4 °C until further use.

### Route II - Immunoprecipitation of p24 antigens from human serum and conversion to Cu ions
Biotin-PEG-CuONPs were linked with avidinylated detection antibodies by mixing for 1 h, and then collected by centrifugation at 3500 x $g$ for 5 min at room temperature. After removal of the supernatant and two additional washes with DI water, the probes were resuspended in 1 mL PBS solution, then blocked with 200 μL BSA (10% in PBS) for 30 min, and stored at 4 °C until further use. The immunoprecipitation protocol in Route II is the same as in Route I, except that detection antibodies were labeled with biotin-avidin linked CuONPs for amplification of the conversion.

### Preparation of DNA-AA@CB[6] probes
Conjugation between DNA-Alkyne (sequence: 5'-CCCCCCCCCCT* CCCCCCCCCC-3', T* indicates alkyne-modified thymine; Sangon Biotechnology Co. Ltd.) and AA was carried out using Cu+ catalyzed click chemistry. In general, 3 μL DI water, 4 μL HEPES (100 mM) buffer, 3 μL alkyne-functionalized DNA (100 μM in DI water), 4 μL AA (200 mM in acetonitrile), 3 μL ascorbic acid (20 mM), were mixed with 4 μL Cu2+ eluent from immunoprecipitation assays (40 mM copper nitrate for characterization samples). The reaction was incubated on a shaker for 4 h at room temperature before termination by adding 4 μL EDTA solution (100 mM). The product (DNA-AA) was purified in batches by centrifugation using Micro Bio-spin P6 columns. Finally, 15 μL CB[6] aqueous solution (5 mM) was added to DNA-AA and incubated for 4 h. The resulted DNA-AA@CB[6] probe solution was stored at 4 °C for further use.

### Kinetic study of the Cu+ ion catalyzed click reaction
Different Cu+ ion concentrations (0–10 mM) and various reaction times (10–720 min) were adopted to investigate the click reaction

efficiency. The reaction was performed according to the above-mentioned protocol. Final reaction mixtures were subjected to mass spectrometry analysis before incubation with CB[6]. Concentrations of species were determined by peak area integration.

### Clinical samples

Serum samples and associated clinical data were collected using a University of South Carolina IRB approved protocol from enrolled adults who visited Prisma Health Richland Hospital from March 2020 to October 2021 for medical evaluation. All study subjects were evaluated by clinicians in the Prisma Health Immunology Center and were enrolled only after written informed consent was obtained. No statistical methods were used to pre-determine sample size, which was determined by the number of patients agreed to participate during the study period. Samples were excluded if there were possible contamination, known COVID-19 infection, or missing information. Among the 118 eligible subjects (age 20–78), 86 (73%) are male. Sample classification was based on clinical diagnosis and viral load. No further selection has been made to either positive or negative samples. All clinical samples were collected using serum separator tubes. Serum in each tube was collected after the formation of clot and aliquoted before storage at −80 °C.

### Single-channel current recording

Fabrication of α-HL nanopore sensors follows a traditional method previously reported[84]. Briefly, 1,2-Diphytanoyl-sn-glycero-3-phosphocholine was used for self-assembly of a synthetic lipid bilayer across an aperture (diameter: 200 μm) on a 25 μm-thick Delrin wall that divided a planar bilayer chamber into two compartments: cis and trans. Both compartments contained 1 mL of work solution. Electrical potential was applied to the *trans* side using Ag/AgCl electrodes with a Planar Lipid Bilayer Workstation (Warner Instruments) and slowly ramped up to examine the stability of the membrane at ±200 mV. The membrane capacitance was maintained at 160–180 pF with various voltage bias values throughout each experiment. A small amount (-0.05 μg) of α-HL protein were added to the *cis* compartment while the trans voltage was changed to +100 mV to drive a single protein into the lipid bilayer. After a stable α-HL protein was inserted and confirmed by an open nanopore current, analytes were added to the *cis* chamber (grounded), and the ionic current through the pore was recorded under a bias of 160 mV. All experiments were carried out at 25 ± 2 °C.

### Data collection and analysis

Ionic current recordings were collected using a patch clamp amplifier (Clampex, version 11.0.3, Warner Instruments) with a built-in high-pass filter with a corner frequency of 5 kHz. Signals were digitized by a Digidata 1440 A analog-to-digital converter (Molecular Devices) at a sampling frequency of 100 kHz and processed by pClamp software (version 11.0, Molecular Devices). Each sample was measured at least three times using independent nanopores. The raw data was analyzed using an in-house MATLAB (Version R2019a) based algorithm to find the current blockade and the dwell time of each eligible signal event, which are two commonly used parameters for identifying different analytes. The current blockade (i.e. residual current) that represents the capture of single molecules and their translocation through the nanopore is defined as $I/I_O$ ($I$: average current measured with the analyte inside the pore; $I_O$: the average baseline value in the absence of analytes). Dwell time (i.e. duration) represents the effective interaction time between the nanopore and a single analyst. For the quantification of biomarkers, the frequency of multi-level signature events generated by translocation of probe DNAs were determined by manual inspection to the raw data. Clampfit (version10.7), OriginPro (version 9.0), Graphpad Prism (version 9.0), were used for data analyzing, histogram construction, curve fitting,

and graph presentation. Python (version 3.7) modules Matplotlib and Seaborn's bivariate kernel density estimator were used for scatter plots and contour plots.

### Statistics and reproducibility

For each of the nanopore-related results, the number of replicates or statistic events was mentioned in the corresponding figure caption or supplementary tables. Recorded current traces with complete blocking of the nanopore ($I/I_O = 1$) were excluded from the statistics for accurate signal capture rate calculation. No data were further excluded from the analysis. Clinical information was blinded until all experiments and p24 quantification was completed.

### Reporting summary

Further information on research design is available in the Nature Research Reporting Summary linked to this article.

## Data availability

The main data supporting the findings of this study are available within the article, the Supplementary Information file, and the Source Data file. The raw datasets generated during the study are not publicly shared but are available for research purposes from the corresponding author upon reasonable request. Source data are provided with this paper.

## Code availability

Source code used in this work is available for research purposes from the corresponding author upon reasonable request.

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

## Acknowledgements

We thank Drs. Qian Wang, Michael Walla, and William Cotham for their invaluable guidance in material characterization. We acknowledge supports from: the National Institute of Allergy and Infectious Diseases (NIAID) grant K22AI136686 to C.L., the South Carolina IDeA Networks of Biomedical Research Excellence Developmental Research Project grant to C.L. funded by the National Institute of General Medical Sciences (NIGMS) grant P20GM103499, and the National Science Foundation (NSF) CAREER award 2047503 to C.L. The characterization work in this study was supported by the NSF Major Research Instrumentation (MRI) grant 1828059 for the acquisition of a Thermo Q-Exactive mass spectrometer. The authors are grateful to the study participants and the medical professionals at the Prisma Health Richland Hospital.

## Author contributions

X.W. and C.L. conceived the project and designed the experiments. C.L. managed the project. X.W. performed single-channel recording experiments and characterization experiments and collected the data. X.Q.W. and Z.Z. supported single-channel recording experiments and clinical sample processing. H.A. collected clinical samples and information. Y.L, Z.W., and W.X. performed characterization experiments of probes. J.M., H.W., and C.T. supported characterization experiments. P.J., T.H., and H.A. analyzed the data. X.W. and C.L. analyzed the data and wrote the paper. All authors participated in drafting and revising the manuscript.

## Competing interests

C.L has a provisional patent "Click Chemistry Amplified Nanopore Assay for Ultrasensitive Quantification of Proteins" submitted through the University of South Carolina. Other authors declare no competing interests.
