## [Peer Review File · Nature Communications]

Reviewers' Comments:

Reviewer #1:

Remarks to the Author:

The manuscript by Wei, et al. describes a strategy for the ultrasensitive detection of p24 antigen, an early virological biomarker of HIV-1 infection. They employ a click chemistry amplified nanopore (CAN) assay through which differentiable DNA probes are formed via a Cu ion catalyzed azide-alkyne click reaction and then quantified by alpha hemolysin translocations. When compared with copper nanocluster-based immunoassays and ELISA, their assay exhibits 20-100x higher analytical sensitivity, allowing superior detection of p24 in human serum samples (down to 10 pg/mL).

While the presented assay is more complex and less time-efficient than e.g. ELISA, the major outcome of the manuscript is the improved limit of detection that would enable more sensitive (i.e. earlier or more trace) detection of the important bioindicator. The methods are described sufficiently and the data is presented clearly. For these reasons, the paper is of sufficient quality and impact to be published in Nature Communications. However, I feel there are several points that should be addressed prior to acceptance, as described below.

1. The current manuscript presents the CAN assay as a novel approach, but in fact it has been reported previously in its entirety in DOI: 10.1002/smll.201804078, featuring at least one of the present authors. While this paper was cited (ref. 31), it was only in passing, never making it clear that the current work is actually an extension of that past report. I found this to be misleading in that it suggests a higher level of novelty than actually exists in the work. In actuality, the only assay innovation here is the use of an alternative Cu-carrying system that yields four Cu ions instead of one for each target protein. This needs to be made much clearer.
2. The authors state in the introduction section: "To ensure high specificity, the DNA probe is specially designed to induce a unique signal that is clearly different from translocation signals of any other known biomolecules." This is an overstatement (the authors cannot have tested against all other known biomolecules) and should be revised accordingly. At any rate, due to the specificity of capture and the fact that measurements are not performed in the complex biofluid itself, it really shouldn't matter if it is completely unique, just that it can be differentiated from other molecules that could reasonably be in solution.
3. Is there any specific significance to the DNA sequence used other than ensuring no secondary structure in the probe?
4. Do the authors observe any translocations for CB[6] alone, given that it will be present in the measurement solution? If not, would it be possible to provide more CB[6] and shorten the incubation time for the DNA-AA@CB[6] formation?
5. In Fig. 3f (and elsewhere), the use of ng/mL is less informative than plotting in nM. Unless the authors expect there to be any dependence of the assay on the molecular weight of the target protein? At the least, the authors could provide a top axis on all pertinent plots showing the same values in nM. This also will allow easier comparison to the past report referenced above.
6. The data in Fig. S5 showing translocation frequency of DNA-AA@CB[6] decreases significantly with salt gradient is in opposition to past reports (e.g. DOI:10.1063/1.4855075). When a higher salt concentration is used on the trans side, the event rate increases. Please explain this contradictory behavior. Is it related to dissociation of construct/complex?
7. Provide any characterization done on biotin-PEG-CuONPs and DNA-AA@CB[6].
8. Please explain why healthy donor serum was diluted with buffer (1:1 ratio) before spiking any P24 antigen. Was this to increase antigen-antibody binding efficiency?

Smaller grammatical errors:

Figure 1: adamantane is misspelled.

Pg. S5: "until the mixture changed toclear" should read "until the mixture changed to clear"

Reviewer #2:

Remarks to the Author:

Wei et al have described an interesting ultrasensitive approach to sensing HIV p24 antigen in clinical samples using click chemistry to produce modified DNA-AA@CB complexes that provide specific signatures when translocating aHL, a protein-based biological nanopore. It is clear that they have done a lot of work, and that they have performed many characterization experiments to demonstrate the utility of their assay and sensor, as well as assess quantitative clinical validity of p24 detection in modest sized samples. Comparing to their recently published previous work, it appears that for this work, the authors have expanded their sandwich assay from one that used the preassembled DNA-AA@CB attached to gold nanoparticles for detection and heat denaturation prior to nanopore-based signal generation, to a system that recovers Cu ions to perform the necessary click chemistry downstream of the initial molecular recognition event. The authors claim this strategy demonstrates improved LOD and specificity compared to other methods.

There are a few major issues with the manuscript in its present form, which are summarized below, that lead to the recommendation to reject the submission for this journal. Also summarized below are some more minor suggestions which the authors may find useful before submitting their manuscript elsewhere.

Major Comments:

1) The general sensing schematic is not explained well enough for the reader to understand the advantages of the method. Though the introduction provides clear motivation for the development of the CAN assay, the reader would benefit from more specific justification as to why each component of the assay is necessary. Likewise in the discussion it would be helpful to explain why this workflow is better than their simpler approach very recently described by Zhang et al., in *Biosensors & Bioelectronics* 181 (2021) 113134, which is only briefly referenced as an example in the introduction (this previous work by the same group affects the novelty of this manuscript). Specifically, the strategy for signal amplification via the recovery and use of Cu ions is not clear, nor what amplification factors are achieved. Do the CuO particles contain a single Cu ion (that translates to a 4X amplification) or do they contain multiple ions which should facilitate amplification over several orders of magnitude. The mechanism is presented differently in two figures, one suggests Cu²⁺ (Figure 1) is necessary for the click chemistry and the other Cu⁺ (Fig3a). Please clarify. Finally, it is not clear whether the click reaction is concentration or time dependent and how this may affect the formation of the final sensing complex. While, a calibration curve for the ratio of converted DNA into DNA-AA versus Cu ion concentration is shown in Figure 3b, it is not clear if that range of Cu ion concentration is relevant for the range p24 protein studied because the amplification factor is not discussed. This raises additional questions such as: is CB[6] binding to all DNA-AA? How does equilibrium of the DNA-AA@CB[6] complex affect sensing? How efficient is this reaction? if the reaction does not go to completion or equilibrium, what is the distribution of subpopulations of DNA, DNA-AA and DNA-AA-CB[6] in a sample for different Cu ion concentrations? How is it analyzed to calculate the capture rate?

2) There are two instances in this work where there seem to be flaws in the data analyses. The first is Figure 3f where the authors claim there is an "excellent" linear correlation between multi-level signature event frequency and p24 concentration. This conclusion is erroneous. Since both scales are log, to maintain a linear relationship the order of magnitude for each corresponding x-y pair would need to be the same which is not the case, therefore invalidating the conclusion. This data should be reanalyzed and presented alongside the sample size or number of events for each point, and a clear explanation of where the error bars/standard deviation are derived from should be included in the caption (all this important info is missing).

3) The lack of precision in the data is also concerning. There is overlap of the error bars between points, so that it is not clear if the method can distinguish a 10x change in concentration in the

protein target. This is not discussed. Overall, there is too much emphasis on the sensitivity performance (LOD), while other aspects of the assay are not addressed, including, precision, dynamic range, time. From the data presented in Fig 3f, the precision is poor and the capture rate is very slow (events/min would lead to a very long analysis time to collect sufficient population data, especially at low protein target concentration, where a low fraction of DNA molecules are expected to be converted to DNA-AA-CB[6]), which would lead to long measurement time to detect enough DNA-AA-CB[6] to calculate a capture rate). It is also unclear as to how capture rate for DNA-AA-CB[6] subpopulation was determined. Were the multiple populations distinguished before capture rate was calculated? How many events were counted? These analyses are not trivial and the process by which they are performed should be described and discussed as this will affect the error bar and thus the precision.

4) Considering the above comments, the LOD quoted of 0.5pg/mL is very questionable, making the claim of a more sensitive assay than commercial ELISA by >10x also questionable.

5) The second instance in this work where the analysis was flawed was in figure 5. The authors should justify why student's t-test was used for pairwise comparison rather than an ANOVA with post-hoc analyses (such as Tukey), which would have allowed for comparison between multiple groups. This would apply to the data shown in figure 5d and e. From the information provided it seems like an ANOVA would have been a more appropriate statistical measure. The caption is also missing important details. Please edit the caption to include all of the relevant statistical information including sample size and test values. Also consider editing the graph to more clearly indicate which groups are being compared when the bar spans across three samples. These major comments question the novelty of the work, but also (and maybe more importantly) the validity of the conclusions so that publication in Nature Communication cannot be recommended and the authors should carefully review and address these comments before resubmitting their work to another journal.

Minor Comments:

1) A lot of work was done to compare the method to existing/alternative methodologies. It would benefit the reader to include a table comparing the methods either in the main text or supplemental (or to include these data in table S1).

2) There are a few small typos throughout the manuscript that should be corrected, for example figures 3b and S6 are labeled "abumdance" rather than "abundance"

3) There are several instances in the text where the data is not presented objectively. The manuscript should be carefully reviewed to ensure the information is presented objectively. One example includes where the authors claim that their method demonstrates a 20-100 fold higher analytical sensitivity than existing methods yet the LOD reported is 0.5 pg/mL for the CAN assay and the LOD reported for a commercial ELISA kit: <https://www.abcam.com/hiv1-p24-elisa-kit-ab218268.html> is 1 pg/mL.

4) Please carefully examine Figure 2. I think there may be an error in the DNA structure shown in figure 2g, it looks like unmodified DNA, but is this trace showing alkyne modified DNA? Please clarify. If possible, please lower the red triangles down from the trace in 2e so it is easier to see the 2-level trend when comparing the three distinct traces.

5) Page 9 line 3: please check review the use of the word significant and its other forms without including the specific statistical values associated with any analysis performed. This would also apply to page 16 lines 6 and 7. For example, the authors may be required to include data as follows in the example "there was a significant increase in the positive versus control group ($F(2,27) = 1.4, p=0.1$)..."

6) The way figure 3 d is coloured currently it looks like the legend only relates to the Fe ions.

7) Figure 5 a is slightly confusing, some of the arrows seem to indicate contrary information, for example the clinical diagnosis: HIV panel points to both the HIV and non-HIV controls.

8) It is not clear why the COVID-19 patients were included? The group is not big enough to draw statistically valid comparison, and its presence is not discussed past the figure. Consider removing these data points until more samples are available for analysis.

9) On page 5 there are several statements that need to be corrected

a. Line 1: The authors state that "By measurement and statistical analysis [...] via the frequency of the blockade events". Maybe clarify that modified DNA molecules must be in $> \text{nM}$ for this to work or else it takes too long to capture a statistically significant number of molecules.

b. Line 3-4: The authors state that "However, due to [...] towards any analytes", however there are examples of aHL nanopores distinguishing SNPs, 3' to 5' orientation, etc. This statement should be revised.

c. Line 7: the authors state "[...] not suitable for detecting electrically neutral molecules [...]", yet many examples exist of such molecules being tested. Further not all proteins and antibodies are neutral, many have charges and thus a dipole moment. Please correct this statement and add appropriate references.

10) On Page 8, the authors mention limiting their applied voltage to limit membrane capacitance. I do not understand this statement as their capacitance should depend on the area, thickness and dielectric properties of membrane but not on voltage.

11) On page 8, significant figures for blockages should match the error reported.

Response to Reviewers' Comments on NCOMMS-21-13368B-Z

Dear Dr. Ross Cloney,

We greatly appreciate yours and Dr. Kyle Legate's efforts in reviewing our manuscript, and also thank all the reviewers for their constructive and thoughtful comments. In this revised version of the manuscript, we believe that we have addressed all points raised by the reviewers through additional experiments, analysis, and discussion, which have significantly strengthened the work. Here, we highlight the key improvements and revisions to the manuscript:

- 1) As suggested by Reviewer 1, we have better clarified the innovation of the current work, changed the unit of p24 concentration, and performed additional experiments to characterize structures of biotin-PEG-CuO nanoparticles and DNA-AA@CB[6] probes and the effect of CB[6] incubation time on DNA-AA@CB[6] formation.
- 2) To address Reviewer 2's concerns on the amplification factor and the click reaction, we have performed additional experiments to characterize the Cu ion catalytic click reaction and additional analysis on the amplification factor along with theoretical calculation. We have also and improved the precision of data and the statistical method for comparative analysis between multiple groups.
- 3) A more rigorous clinical evaluation of the CAN assay was performed through blinded testing in an expanded cohort.

With this revision, we find the science in this manuscript significantly improved and strengthened. We believe that the platform technology for sensitive proteomic biomarker testing described in this paper will be of great interest to the clinical diagnostics field. Below is a point-by-point response to each comment from the reviewers (under "RE:" in blue). Text added into the manuscript is shown in orange.

Response to Reviewer #1

The manuscript by Wei, et al. describes a strategy for the ultrasensitive detection of p24 antigen, an early virological biomarker of HIV-1 infection. They employ a click chemistry amplified nanopore (CAN) assay through which differentiable DNA probes are formed via a Cu ion catalyzed azide-alkyne click reaction and then quantified by alpha hemolysin translocations. When compared with copper nanocluster-based immunoassays and ELISA, their assay exhibits 20-100x higher analytical sensitivity, allowing superior detection of p24 in human serum samples (down to 10 pg/mL). While the presented assay is more complex and less time-efficient than e. g. ELISA, the major outcome of the manuscript is the improved limit of detection that would enable more sensitive (i.e. earlier or more trace) detection of the important bioindicator. The methods are described sufficiently and the data is presented clearly. For these reasons, the paper is of sufficient quality and impact to be published in Nature Communications. However, I feel there are several points that should be addressed prior to acceptance, as described below.

RE: We are grateful for the reviewer's positive summary, comments, and suggestions on our work. We now believe that we have addressed all concerns from the reviewer, which resulted in a significantly improved manuscript. A point-by-point report below lists our response to each question/suggestion for reviewer's reference.

1. The current manuscript presents the CAN assay as a novel approach, but in fact it has been reported previously in its entirety in DOI: 10.1002/sml.201804078, featuring at least one of the present authors. While this paper was cited (ref. 31), it was only in passing, never making it clear that the current work is actually an extension of that past report. I found this to be misleading in that it suggests a higher level of novelty than actually exists in the work. In actuality, the only assay innovation here is the use of an alternative Cu-carrying system that yields four Cu ions instead of one for each target protein. This needs to be made much clearer.

RE: We thank the Reviewer for raising this important issue. We regret for any misleading information. Following the reviewer's suggestion, we have revised the Introduction (Page 4-5) to appropriately reflect the innovation and clinical impact of the present work. In addition to the innovation on the Cu-carrying system recognized by the reviewer, we would like to summarize several other new understandings reflected in this manuscript. The previous paper (DOI: 10.1002/sml.201804078) reports a standard curve constructed in buffer. This experimental flaw leads to unreliable quantification of biomarkers and false claim of detection limit and sensitivity in human serum. Our new manuscript presents a standard curve made in the same experimental condition as in human serum testing. The new manuscript also presents step-by-step quantitative characterization results (MS, NMR, FTIR, etc.) of the chemical reactions in the assay to ensure highest accuracy and reproducibility of the method. Finally, only a small fraction of published technical breakthroughs in biosensing advance to clinical diagnostics. This is largely due to the lack of rigorous clinical evaluations to help readers with clinical background gauge the translational value of the reported technologies. The present manuscript fills this gap with a clinical performance assessment conducted in a well-characterized HIV cohort.

“Furthermore, quantitative chemical modifications to DNA probes with concentration dependent probe yield and highly characteristic current blockade events were utilized in nanopore sensing to improve sensitivity and recognizability of the desired signals.³⁵ For example, by incorporating a sandwich assay involving copper oxide nanoparticles, a host-guest modified DNA probe catalyzed by assay released Cu ions was used to derive concentrations of cancer biomarkers.³⁶ Our group recently used two DNA structures with different modifications as detection reporters for multiplex quantification of immunoglobulin M and G antibodies against the nucleocapsid protein of SARS-CoV-2 in serum specimens from early stage COVID-19 patients.³⁷”

The above-mentioned studies have made great progress in several aspects in developing DNA probe assisted nanopore sensing for various biomarkers with high selectivity and sensitivity. However, many challenges remain ahead towards the goal of a universal biosensing platform. First, a robust amplification strategy is required to further improve the detection sensitivity and achieve a lower LOD for early and accurate detection. Second, relationships between DNA probes, the biomarker, the catalyst, and the host-guest structure in the modification process, as well as the distribution of their respective nanopore signals remain ill-defined, thus hinders further improvements on sensing accuracy and reproducibility. Third, a rigorous clinical performance evaluation is currently lacking, which is of utmost importance in the last mile from bench to bedside.”

2. The authors state in the introduction section: “To ensure high specificity, the DNA probe is specially designed to induce a unique signal that is clearly different from translocation signals of any other known biomolecules.” This is an overstatement (the authors cannot have tested against all other known biomolecules) and should be revised accordingly. At any rate, due to the specificity of capture and the fact that measurements are not performed in the complex biofluid itself, it really shouldn’t matter if it is completely unique, just that it can be differentiated from other molecules that could reasonably be in solution.

RE: We agree with the Reviewer’s suggestion. We have revised this sentence (Page 6) as follows:

“To ensure high specificity, the DNA probe is designed to induce a unique translocation signal that is clearly different from signals of other molecules in the assay system.”

3. Is there any specific significance to the DNA sequence used other than ensuring no secondary structure in the probe?

RE: As the Reviewer mentioned, the most important concern of this sequence (5'-CCCCCCCCCT*CCCCCCCC-3', T* indicates alkyne-modified thymine) is to prevent the formation of secondary structure. In addition, from a synthesis point of view, synthesizing a single base sequence is relatively simple and efficient.

4. Do the authors observe any translocations for CB[6] alone, given that it will be present in the measurement solution? If not, would it be possible to provide more CB[6] and shorten the

incubation time for the DNA-AA@CB[6] formation?

RE: We thank reviewer for this comment. The amount of CB[6] used to incubate with DNA-AA probes is in excess in our protocol. No multi-level signal was observed when CB[6] molecules interact with the α -HL nanopore. Blocking signals observed at high-frequency indicate that it is difficult for free CB[6] molecules to enter the cavity of the nanopore. This result is included in the Supporting Information file (Figure S7).

Inspired by the reviewer's suggestion, we further optimized the incubation time of CB[6] and DNA-AA, and found that the frequency of multi-level signature events reached a plateau after 4 h incubation time, and remained within the same level for 6 h and 12 h incubation time. This result is included in the Supporting Information file (Figure S9b). We have also updated the main text in the Results (Page 11) and the Methods sections (Page 25) to reflect this improvement.

Figure S7: Representative time scaled (20 s and 2 s) raw current traces of CB[6] translocations. **Figure S9b:** multi-level signature event frequency of DNA-AA@CB[6] probes through nanopore as a function of the incubation time with CB[6].

5. In Fig. 3f (and elsewhere), the use of ng/mL is less informative than plotting in nM. Unless the authors expect there to be any dependence of the assay on the molecular weight of the target protein? At the least, the authors could provide a top axis on all pertinent plots showing the same values in nM. This also will allow easier comparison to the past report referenced above.

RE: We thank the reviewer for this suggestion. We have updated the entire manuscript with the use of pM/fM.

6. The data in Fig. S5 showing translocation frequency of DNA-AA@CB[6] decreases significantly with salt gradient is in opposition to past reports (e.g. DOI:10.1063/1.4855075). When a higher salt concentration is used on the trans side, the event rate increases. Please explain this contradictory behavior. Is it related to dissociation of construct/complex?

RE: We acknowledge the Reviewer for this finding. Through statistics analysis of translocation events of DNA probes without CB[6], we find that the frequency of events does increase with increasing KCl concentration in the *trans* side (Figure R1), which is in agreement with mentioned reference. Similar behavior is also observed for other linear macromolecules.

Figure R1. Capture rate of DNA by α -HL nanopore at pH 8.0 for different *cis* and *trans* KCl concentrations.

However, the frequency of multi-level signals is in an opposite trend. Our theory is that well-defined multi-level signals can only be observed under certain translocation speed of DNA-AA@CB[6] probes, which causes oscillation of dissociated CB[6] in the pore lumen. Concentration gradient can increase the “pulling” force and accelerate the dissociation process of DNA-AA@CB[6], thus interfere or prevent the oscillation of dissociated CB[6]. We are currently studying this process using molecular dynamics simulation in another project that focuses on translocation behaviors of various structures.

We have cited the relevant literature and added the following discussion in the revised manuscript (Page 10) as follows:

“Although concentration gradient was found to increase translocation of linear macromolecules,^{45, 46} our results show that optimal multi-level signal frequency can only be achieved using balanced *cis* and *trans* work solutions (*i.e.* with the same electrolyte concentration and pH), and that any disruption of the balance could result in decrease in the frequency (Figure S7).”

7. Provide any characterization done on biotin-PEG-CuO NPs and DNA-AA@CB[6].

RE: We have added more characterization results on biotin-PEG-CuO NPs and DNA-AA@CB[6], including TEM, DLS, FTIR and TGA for biotin-PEG-CuO NPs (Figure 4c, d, f), and NMR for DNA-AA@CB[6] (Figure S9a). Discussion to these results have been added in the revised manuscript (Page 12-14).

In addition, we also added more mass spectrometry results to characterize the DNA-AA click reaction under various conditions (Figure 3), as well as FTIR and TGA results to characterize the synthesis of Cu clusters (Figure S13).

8. Please explain why healthy donor serum was diluted with buffer (1:1 ratio) before spiking any P24 antigen. Was this to increase antigen-antibody binding efficiency?

RE: Yes. The purpose of the dilution is indeed to facilitate antigen-antibody binding. In addition to this reason, we commonly observe lipid-like contaminations/suspensions in some serum samples in clinical HIV cohorts. A 1:1 dilution can help reducing the interference from the suspension while minimizing the impact on detection sensitivity.

Smaller grammatical errors:

1. Figure 1: adamantane is misspelled.

RE: We thank the reviewer for pointing out this error. We have corrected this error in the revised manuscript.

2. Pg. S5: “until the mixture changed toclear” should read “until the mixture changed to clear”

RE: We thank the reviewer for pointing out this error. We have corrected this error in the revised manuscript.

Response to Reviewer #2

Wei et al have described an interesting ultrasensitive approach to sensing HIV p24 antigen in clinical samples using click chemistry to produce modified DNA-AA@CB complexes that provide specific signatures when translocating α -HL, a protein-based biological nanopore. It is clear that they have done a lot of work, and that they have performed many characterization experiments to demonstrate the utility of their assay and sensor, as well as assess quantitative clinical validity of p24 detection in modest sized samples. Comparing to their recently published previous work, it appears that for this work, the authors have expanded their sandwich assay from one that used the preassembled DNA-AA@CB attached to gold nanoparticles for detection and heat denaturation prior to nanopore-based signal generation, to a system that recovers Cu ions to perform the necessary click chemistry downstream of the initial molecular recognition event. The authors claim this strategy demonstrates improved LOD and specificity compared to other methods. There are a few major issues with the manuscript in its present form, which are summarized below, that lead to the recommendation to reject the submission for this journal. Also summarized below are some more minor suggestions which the authors may find useful before submitting their manuscript elsewhere.

RE: We are extremely grateful for the Reviewer's comments and criticisms to help us improve this assay. Following the Reviewer's suggestions, we have added a significant number of experiments and discussion to address key scientific issues. Meanwhile, we have also included additional clinical results collected during the revision period to demonstrate the assay's robust clinical performance. Collectively, we now believe that the current manuscript reports a highly rigorous study of the assay's innovation, mechanism, reproducibility, and clinical significance.

Major Comments:

1. The general sensing schematic is not explained well enough for the reader to understand the advantages of the method. Though the introduction provides clear motivation for the development of the CAN assay, the reader would benefit from more specific justification as to why each component of the assay is necessary. Likewise in the discussion it would be helpful to explain why this workflow is better than their simpler approach very recently described by Zhang et al., in *Biosensors & Bioelectronics* 181 (2021) 113134, which is only briefly referenced as an example in the introduction (this previous work by the same group affects the novelty of this manuscript). Specifically, the strategy for signal amplification via the recovery and use of Cu ions is not clear, nor what amplification factors are achieved. Do the CuO particles contain a single Cu ion (that translates to a 4X amplification) or do they contain multiple ions which should facilitate amplification over several orders of magnitude. The mechanism is presented differently in two figures, one suggests Cu^{2+} (Figure 1) is necessary for the click chemistry and the other Cu^+ (Fig3a). Please clarify. Finally, it is not clear whether the click reaction is concentration or time dependent and how this may affect the formation of the final sensing complex. While a calibration curve for the ratio of converted DNA into DNA-AA versus Cu ion concentration is shown in Figure 3b, it is not clear if that range of Cu ion concentration is relevant for the range p24 protein studied because the amplification factor is not discussed. This raises additional questions such as: is CB[6] binding to all DNA-AA? How does equilibrium of the DNA-AA@CB[6] complex affect sensing?

How efficient is this reaction? if the reaction does not go to completion or equilibrium, what is the distribution of subpopulations of DNA, DNA-AA and DNA-AA-CB[6] in a sample for different Cu ion concentrations? How is it analyzed to calculate the capture rate?

RE: We sincerely acknowledge the Reviewer for the above comments. We have broken down these comments to several issues. Below we will respond to these issues point by point.

- 1.1** The general sensing schematic is not explained well enough for the reader to understand the advantages of the method. Though the introduction provides clear motivation for the development of the CAN assay, the reader would benefit from more specific justification as to why each component of the assay is necessary. Likewise in the discussion it would be helpful to explain why this workflow is better than their simpler approach very recently described by Zhang et al., in *Biosensors & Bioelectronics* 181 (2021) 113134, which is only briefly referenced as an example in the introduction (this previous work by the same group affects the novelty of this manuscript).

RE: We acknowledge the Reviewer's suggestion, and have revised Figure 1 and the Introduction section to better explain and justify the sensing schematic.

Firstly, we have introduced several related previous works in details in the Introduction, and have put forward the challenges faced by these previous work as follows, which we aimed to overcome in the current study (Page 4-5):

“Furthermore, quantitative chemical modifications to DNA probes with concentration dependent probe yield and highly characteristic current blockade events were utilized in nanopore sensing to improve sensitivity and recognizability of the desired signals.³⁵ For example, by incorporating a sandwich assay involving copper oxide nanoparticles, a host-guest modified DNA probe catalyzed by assay released Cu ions was used to derive concentrations of cancer biomarkers.³⁶ Our group recently used two DNA structures with different modifications as detection reporters for multiplex quantification of immunoglobulin M and G antibodies against the nucleocapsid protein of SARS-CoV-2 in serum specimens from early stage COVID-19 patients.³⁷

The above-mentioned studies have made great progress in several aspects in developing DNA probe assisted nanopore sensing for various biomarkers with high selectivity and sensitivity. However, many challenges remain ahead towards the goal of a universal biosensing platform. First, a robust amplification strategy is required to further improve the detection sensitivity and achieve a lower LOD for early and accurate detection. Second, relationships between DNA probes, the biomarker, the catalyst, and the host-guest structure in the modification process, as well as the distribution of their respective nanopore signals remain ill-defined, thus hinders further improvements on sensing accuracy and reproducibility. Third, a rigorous clinical performance evaluation is currently lacking, which is of utmost importance in the last mile from bench to bedside.”

Secondly, we optimized Figure 1 and modified corresponding descriptions in the caption (page 6)

for clarity:

Figure 1. Schematic illustration of the click chemistry amplified nanopore (CAN) assay for quantification of HIV-1 p24 antigen in human serum: (1) blood samples were collected into serum separator tubes, and sera were collected after centrifugation and clot formation; (2) serum samples were incubated with capture antibody-modified magnetic beads (MBs) and detection antibody-modified copper oxide nanoparticles (CuONPs) to form a sandwich structure between MBs, enriched p24 antigens and CuONPs; (3) sandwich complexes were magnetically separated, and Cu⁺ ions were released from them under acidic conditions; (4) DNA probes were formed by a Cu⁺ ion catalyzed click reaction and a modification with a pair of host-guest molecules; (5) finally, DNA probes were collected and subjected to single-channel recordings using a α -hemolysin (α -HL) nanopore for p24 antigen quantification. The illustration is not drawn to scale. (Created with BioRender.com)

Finally, we added discussion on the mentioned previous work (Page 5-6) as follows to better explain advantages of the current method:

“Comparing to our previous DNA-assisted nanopore sensor for SARS-CoV-2 antibodies,³⁷ in which detection solely depends on separation by immunosorbent beads and conversion to DNA probes preloaded on gold nanoparticles together with detection antibodies, the present CAN assay has two build-in amplifications by catalytic click reaction and biotin-avidin binding to achieve three orders of magnitude lower LOD, allowing early detection of antigens in sera.”

1.2 Specifically, the strategy for signal amplification via the recovery and use of Cu ions is not clear, nor what amplification factors are achieved. Do the CuO particles contain a single Cu ion (that translates to a 4X amplification) or do they contain multiple ions which should facilitate amplification over several orders of magnitude. The mechanism is presented differently in two figures, one suggests Cu²⁺ (Figure 1) is necessary for the click chemistry and the other Cu⁺ (Fig 3a). Please clarify.

RE: We thank the Reviewer for this comment. We have modified multiple sections and figures throughout the manuscript to clarify the signal amplification strategy and related factors. Briefly, the ratio of p24 antigen to CuO particle was first enlarged from 1:1 to 1:4 through biotin-avidin binding (Figure 4a-c); then the yield of DNA probes is amplified by the click reaction catalyzed by Cu ions released from CuO particle, each CuO nanoparticle (~50 nm) can release multiple Cu ions; signature signals of DNA probes were quantified by utilizing nanopore’s single molecule

sensitivity towards DNA.

The mechanism of the Cu ion catalyzed click reaction has been reported in a number of publications (*Chemical Society Reviews*, 36(10), 1674-1689). While Cu^{2+} ions were first released from CuO nanoparticles by HCl, it was Cu^+ ions that catalyzed the click reaction between alkyne modified DNA and 1-Azidoadamantane. Sodium ascorbate was used to reduce the divalent Cu ions to monovalent Cu ions. Therefore, in the original manuscript, Cu^{2+} was shown in sensing schematic in Figure 1, and Cu^+ was shown in the reaction mechanism in Figure 3. To avoid any confusion, we have stressed the reduction process in the revised manuscript (Page 10-12), and have updated figures using the term “Cu ions”.

- 1.3** Finally, it is not clear whether the click reaction is concentration or time dependent and how this may affect the formation of the final sensing complex. While a calibration curve for the ratio of converted DNA into DNA-AA versus Cu ion concentration is shown in Figure 3b, it is not clear if that range of Cu ion concentration is relevant for the range p24 protein studied because the amplification factor is not discussed. This raises additional questions such as: is CB[6] binding to all DNA-AA? How does equilibrium of the DNA-AA@CB[6] complex affect sensing? How efficient is this reaction? if the reaction does not go to completion or equilibrium, what is the distribution of subpopulations of DNA, DNA-AA and DNA-AA-CB[6] in a sample for different Cu ion concentrations? How is it analyzed to calculate the capture rate?

RE: We sincerely appreciate the Reviewer for raising these points, which were indeed not clear in the original submission. To address Reviewer’s concerns and to further improve the quality of the study, we have systematically investigated this reaction process and substantially revised the manuscript. Corresponding results were summarized in Figure 3 and Figure S8-S10 in the revised manuscript accompanied with relevant discussions (Page 11-13).

- 1.3.1** Whether the click reaction is concentration or time dependent and how this may affect the formation of the final sensing complex?

RE: We performed the reaction with different Cu^+ ion concentrations and different reaction times, characterized the products by mass spectrometry (MS), and calculated the reaction efficiency under each condition.

Our new results clarify that the click reaction in our protocol is dependent on the concentration of Cu^+ ions. No DNA-AA signal can be detected by mass spectrometry in absence of Cu^+ ions in the reaction system, even when the reaction time was extended to 12 h (Figure 3c). However, DNA-AA signal increased when Cu^+ ion concentration was increased from 0 to 10 mM (Figure 3d). Although further kinetic study shows that the reaction yield can be affected by reaction time for various concentrations of Cu^+ ions (Figure 3e), our present study used 240 minutes reaction time for all DNA probes. The click reaction was terminated by EDTA to eliminate any reaction time effect on the yield.

Figure 3. **c:** MS characterizations of the substrate and the product of the click reaction under different reaction time in absence of Cu^+ ions. **d:** MS characterizations of the substrate and the product of the click reaction with different Cu^+ ion concentrations under the same reaction time. **e:** Reaction efficiency as a function of reaction time in presence of various Cu^+ ion concentrations. Reaction efficiency is determined by $S_{\text{DNA-AA}}/(S_{\text{DNA}} + S_{\text{DNA-AA}})$ (S : integrated peak area in mass spectrum).

1.3.2 It is not clear if that range of Cu ion concentration is relevant for the range p24 protein studied?

RE: We thank the Reviewer for pointing this out. For the reaction time used (240 minutes), a positive correlation was observed between the reaction efficiency and the Cu^+ ion concentration. **Figure 3f** highlights a standard curve for reaction efficiency (percentage of converted DNA into DNA-AA) vs. Cu^+ ion concentration. Although the reaction efficiency reached a plateau at 10 mM Cu^+ ions, a good linear relationship was obtained in the range of 0-1 mM.

We further calculated corresponding p24 concentrations within this linear interval of Cu^+ ion concentrations. Assuming a 1:1 ratio of p24 antigen and CuO nanoparticle, we found that the theoretical antigen concentration range covered is 0~145 pM (0~3500 pg/mL). After extending this ratio to 1:4 (**Inset of Figure 3f**), the linear range of Cu^+ ion concentrations still covers the p24 range we studied (0~800 pg/mL) in all of our clinical samples.

Next, we examined the nanopore signature signal frequency of DNA-AA@CB[6] synthesized with different concentrations of Cu^+ ions. As illustrated in **Figure 3g**, an excellent linear relationship can be observed in 0-3 mM Cu^+ ions.

Figure 3. **f:** Reaction efficiency as a function of Cu^+ ion concentrations under 4 h reaction time with linear fit between 0 and 1 mM Cu^+ ions. Inset shows the theoretical linear p24 antigen concentration range covered. **g:** Correlation between multi-level signal frequency and Cu^+ ion concentration. Inset shows the correlation in higher range up to 10 mM. Data represents mean \pm SD of three replicates. Solid line indicates linear regression. Shadow indicates limits of 95% confidence interval.

1.3.3 Is CB[6] binding to all DNA-AA? How does equilibrium of the DNA-AA@CB[6] complex affect sensing? How efficient is this reaction?

RE: We thank the reviewer for this comment. The amount of CB[6] used in this protocol is excessive for obtaining maximum DNA-AA@CB[6] complex. Additional NMR characterization was performed to confirm the DNA-AA@CB[6] product obtained after incubation. The binding between CB[6] and DNA-AA probes can be deduced *via* the chemical shift of protons of the CB[6] molecule (Figure S9a). Although it is difficult to quantitatively determine the reaction efficiency between DNA-AA and CB[6] by NMR, we evaluated the effect of incubation time to the reaction equilibrium based on the final nanopore signature. As shown in Figure S9b, the frequency of multi-level signature events reaches a plateau at 4 h incubation time, which is used for all standard and clinical samples in the manuscript for maximum sensitivity.

Figure S9. a: ¹H NMR (400 MHz) spectra of DNA-AA, CB[6] and DNA-AA@CB[6]. **b:** multi-level signature event frequency of DNA-AA@CB[6] probes through nanopore as a function of the incubation time with CB[6]. In Figure b, frequency of the multi-level signature events of DNA-AA@CB[6] was gradually increase with incubation time and reached a plateau at around 4 h.

1.3.4 If the reaction does not go to completion or equilibrium, what is the distribution of subpopulations of DNA, DNA-AA and DNA-AA-CB[6] in a sample for different Cu ion concentrations?

RE: We thank the Reviewer for this comment. With abovementioned results, we know that the click reaction between DNA and AA was terminated by EDTA after a 4 h reaction; and that the interaction between DNA-AA and CB[6] reaches equilibrium in 4 h. Under these conditions, we further investigated the distribution of DNA and DNA-AA for reactions under different Cu⁺ ion concentrations by HPLC. It can be observed from Figure S10 (right) that, with increasing Cu⁺ ion concentration, the DNA abundance decreased while the DNA-AA abundance increased gradually. However, the distribution of DNA and DNA-AA cannot be precisely quantified by HPLC due to its limited sensitivity.

Due to the stochastic nature of nanopore sensing, signals of the same analyte (*i.e.* DNA or DNA-AA) are different from test to test. Thus, the distribution of DNA and DNA-AA subpopulations cannot be quantified by conventional statistics of nanopore signal either (Figure S10, middle). However, we still attempted to mark representative signals of DNA and DNA-AA on nanopore current traces (Figure 2e). Current blockade and dwell time analysis of ≥ 500 events shows poor signal reproducibility that prevents precise quantification (Figure 2i&j). This is the exact motivation to use a simple and unique “yes or no” multi-level signature for biomarker quantification, rather than the complicated yet unreliable statistical analysis.

On the other hand, linear quantitative relationship between multi-level signal frequency and Cu⁺ ion concentration can be easily obtained (Figure S10, left and Figure 3g). By using the multi-level signal frequency as the readout signal, we significantly improved the specificity while preserving

the high sensitivity of nanopore sensing.

Figure S10. Raw current traces in α -HL of DNA-AA@CB[6] probes which obtained with various concentrations of Cu ions (**Left**), and the corresponding two-dimensional scatter plots of DNA-AA translocation signals (**Middle**). Each plot contains ≥ 500 blockade events. Together with the corresponding distribution of subpopulations of DNA and DNA-AA (**Right**) through HPLC.

1.3.5 How is it analyzed to calculate the capture rate?

RE: We thank the Reviewer for this comment. Each raw current trace data recording (≥ 5 mins long) was examined to count the number of multi-level signal events to calculate the capture rate (frequency). Signal events were analyzed using an in-house Matlab algorithm and then confirmed by manual inspection.

2. There are two instances in this work where there seem to be flaws in the data analyses. The first is Figure 3f where the authors claim there is an “excellent” linear correlation between multi-level signature event frequency and p24 concentration. This conclusion is erroneous. Since both scales are log, to maintain a linear relationship the order of magnitude for each corresponding x-y pair would need to be the same which is not the case, therefore invalidating the conclusion. This data should be reanalyzed and presented alongside the sample size or number of events for each point, and a clear explanation of where the error bars/standard deviation are derived from should be included in the caption (all this important info is missing).

RE: We acknowledge the Reviewer for raising this discussion but are hard-pressed by the comment on linear correlation. We would like to clarify that both x-y scales have the same

magnitude after logarithm (Figure 4g, Figure 3f in the original manuscript), thus the claim of linear relationship is valid. This data presentation method was used to better illustrate data points at lower range and has been widely used in previous studies. One typical example is shown below (*Nature communications, 2021, 12(1): 1-12*):

Figure R2. Calibration curves for miR-375-3p (a) and miR-141-3p (b) using asymmetric salt buffer (*Nature communications, 2021, 12(1): 1-12*).

We would like to clarify that each data point on our standard curve is the average result of three parallel tests. The frequency value representing the capture rate of DNA-AA@CB[6] probes in the nanopore was used to quantify p24 concentration. Therefore, regardless of current blockade and dwell time of the signals, we only counted signals with multi-level signature, which has an effective linear correlation with Cu⁺ ion concentration and thus, p24 concentration. We did not use statistical analysis of current blockade and dwell time in biomarker quantifications, thus do not have a number of events present.

For each data point, we prepared three p24 spiked human serum standard samples, and tested them using three different pores. These three frequency results were used to calculate the mean and standard deviation. According to the Reviewer’s suggestion, we have added the explanation of standard deviation in the figure caption as follows:

“Data represents mean ± SD of three replicates”

3. The lack of precision in the data is also concerning. There is overlap of the error bars between points, so that it is not clear if the method can distinguish a 10x change in concentration in the protein target. This is not discussed. Overall, there is too much emphasis on the sensitivity performance (LOD), while other aspects of the assay are not addressed, including, precision, dynamic range, time. From the data presented in Fig 3f, the precision is poor and the capture rate is very slow (events/min would lead to a very long analysis time to collect sufficient population data, especially at low protein target concentration, where a low fraction of DNA molecules are expected to be converted to DNA-AA-CB[6]), which would lead to long measurement time to detect enough DNA-AA-CB[6] to calculate a capture rate). It is also unclear as to how capture rate for DNA-AA-CB[6] subpopulation was determined. Were the multiple populations distinguished before capture rate was calculated? How many events were counted? These analyses are not trivial and the process by which they are performed should be described and discussed as this will affect the error bar and thus the precision.

RE: We thank the Reviewer for the above comments. We have broken down these comments to several issues. Below we will respond to these issues point by point.

3.1 The lack of precision in the data is also concerning. There is overlap of the error bars between points, so that it is not clear if the method can distinguish a 10x change in concentration in the protein target. This is not discussed.

RE: We performed additional parallel experiments for each data point to improve the precision of data. The CAN assay (route II) can now reliably distinguish 10x changes in p24 concentration in human serum (Figure 4g). We also added corresponding discussion in the revised manuscript (Page 14).

Figure 4. g: Correlations between multi-level signature event frequency and p24 concentration in human serum within the range of 0-46.7 pM (0-1000 pg/mL, final concentration: 0, 0.5, 1, 10, 100, 1000 pg/mL) processed using route I and route 2, respectively. Black lines are linear fits to the data points. Insets show the correlations within lower range (0-50 fM). Data represent the means \pm SD of three replicates.

3.2 Overall, there is too much emphasis on the sensitivity performance (LOD), while other aspects of the assay are not addressed, including, precision, dynamic range, time.

RE: We appreciate the Reviewer for this suggestion. We have included more discussion on other aspects of the assay including precision, dynamic range, assay time, etc. in the revised manuscript (Page 17, 21-22) and a summary of previously reported methods for comparison in the Supporting Information (Table S1).

3.3 From the data presented in Fig 3f, the precision is poor and the capture rate is very slow (events/min would lead to a very long analysis time to collect sufficient population data, especially at low protein target concentration, where a low fraction of DNA molecules are expected to be converted to DNA-AA-CB[6]), which would lead to long measurement time to detect enough DNA-AA-CB[6] to calculate a capture rate).

RE: We thank the Reviewer for raising this discussion. Before quantifying p24 in human serum, we have optimized several factors for nanopore detection of DNA-AA-CB[6] probes, including working solution concentration and pH, etc. These electrochemical optimizations ensure that the highest possible frequency of multi-level signals can be obtained. We do understand the Reviewer's concern on the detection time of low concentration samples, in which multi-level signals are captured at a rate between 2 and 4 events/min. We would like to clarify that, instead of using the traditional current blockade and dwell time analysis in an event "population", we only look for the multi-level oscillation signals, which cannot be induced by any other molecules in this

system but the DNA-AA-CB[6] probes. As it is very easy to recognize multi-level oscillation signals from other signals, as illustrated in Figure 2, we do not need to collect a large population for statistical analysis. We found that a 5-minute recording time is sufficient for robust and reproducible measurements of all serum samples, and employed it in this study.

- 3.4 It is also unclear as to how capture rate for DNA-AA-CB[6] subpopulation was determined. Were the multiple populations distinguished before capture rate was calculated? How many events were counted? These analyses are not trivial and the process by which they are performed should be described and discussed as this will affect the error bar and thus the precision.

RE: We thank the Reviewer for this comment. DNA-AA-CB[6] signals were distinguished before the capture rate was calculated based on their unique multi-level signal. For each sample, nanopore current trace was recorded for 5 minutes and the number of multi-level signals was counted to calculate the capture rate, which was then converted to the p24 concentration.

The distribution of probe subpopulations have been described in previous responses. We understand that reviewer is very interested in distinguishing multiple populations during the evolution of DNA-AA-CB[6] probes. However, we would like to point out an important advantage of this assay: it does not rely on statistical analysis of hundreds of events that is used in nanopore sequencing. Instead, it uses the frequency of a highly specific signal generated by ssDNA probes with a host-guest structure to reflect the concentration of p24 antigens. Regardless of high or low current blockade, long or short dwell time, as long as the signal has an oscillation pattern, it will be counted towards the frequency calculation, which will be used to calculate p24 concentrations using the standard curve (Figure 4g).

Due to the single molecule stochastic sensing mechanism of nanopore sensing, it has extremely high sensitivity but poor specificity. When testing the same analyte using the same pore, current blockade and dwell time of the signal may vary from test to test. When testing a mixture of analytes, hundreds to thousands of events are usually recorded to generate a “heat map” on the current blockade vs. dwell time plot. Each analyte should be identifiable as a subpopulation indicated by a peak area on the heat map. However, these peak areas are usually wide and may overlap with each other due to the aforementioned stochastic sensing mechanism, which may lead to false reading. The use of the unique multi-level oscillation signal bypasses the subpopulation analysis and significantly improve the readout specificity. The precision of this readout method has been rigorously tested and demonstrated in the manuscript.

4. Considering the above comments, the LOD quoted of 0.5pg/mL is very questionable, making the claim of a more sensitive assay than commercial ELISA by >10x also questionable.

RE: To address the Reviewer’s concern, we have performed a large number of supplementary experiments as mentioned above. The characterization experiments on the Cu ion catalyzed click reaction (Figure 3) verified the linearity of the crucial Cu ion to DNA-AA conversion in the CAN assay; Repeated experiments were carried out on the standard and clinical serum samples to confirm the data accuracy and reliability. Thus, we believe that the claim of 0.5pg/mL LOD for

the CAN assay is confirmed theoretically and experimentally, and are confident that all results in the manuscript are reliable and reproducible.

It also worth to mention that in order to verify the higher sensitivity of the CAN assay, we conducted a detailed comparison with ELISA in more clinical samples (124 human samples). The results are shown in (Figure 6) in the revised manuscript.

- The second instance in this work where the analysis was flawed was in figure 5. The authors should justify why student's t-test was used for pairwise comparison rather than an ANOVA with post-hoc analyses (such as Tukey), which would have allowed for comparison between multiple groups. This would apply to the data shown in figure 5d and e. From the information provided it seems like an ANOVA would have been a more appropriate statistical measure. The caption is also missing important details. Please edit the caption to include all of the relevant statistical information including sample size and test values. Also consider editing the graph to more clearly indicate which groups are being compared when the bar spans across three samples.

RE: We thank the reviewer for this suggestion. We have used ANOVA with post-hoc analysis for comparison between multiple groups in the revised manuscript (Figure 6). Details of relevant statistical information including sample size and test values are also provided in the figure caption.

Figure 6. Clinical validation in a pilot cohort. **a:** Quantitative measurements of p24 in clinical samples using the CAN assay, with the clinical diagnosis of each patient. The dashed line indicates the LOD of the CAN assay. **b:** Cluster map produced from p24 results measured by the CAN assay and ELISA for 118 eligible patients. **c:** p24 detection accuracy for the CAN assay and ELISA in different patient groups. **d-e:** The average levels of p24 concentrations obtained by (d) the CAN assay and (e) ELISA for different patient groups. **f:** ROC curves by the

CAN assay and ELISA to differentiate HIV and AIDS diagnosis. **g**: Correlation between CD4 counts and CAN measured p24 concentrations. Inset shows densitometric analysis of p24 concentrations across different groups classified by CD4 counts (≤ 350 , 350-500, $\geq 500/\text{mm}^3$). **h**: Correlation between viral loads and CAN measured p24 concentrations. P values in d, e, and g were calculated by one-way ANOVA with post-hoc Tukey tests, **** indicates $P < 0.0001$, * indicates $P < 0.05$, ns indicates no significant difference.

These major comments question the novelty of the work, but also (and maybe more importantly) the validity of the conclusions so that publication in Nature Communication cannot be recommended and the authors should carefully review and address these comments before resubmitting their work to another journal.

RE: We sincerely appreciate the Reviewer's constructive criticism and suggestions. With all abovementioned efforts, we believe that we have successfully addressed all points raised by the Reviewer.

Firstly, we reviewed previous publications and summarized challenges faced by this field. We then re-elaborated the novelty of the CAN assay and the new understanding this study presents to the field (Page 4-5). Secondly, we carried out detailed characterization studies for each important chemical reaction in the CAN assay, which allow us to establish a robust standard curve for p24 quantification. Thirdly, we experimentally compared the CAN assay with existing sensing strategies side-by-side, including fluorescent Cu nanocluster assay and ELISA, and confirmed the higher sensitivity of the CAN assay for p24 detection in serum samples. We also compared the CAN assay with other p24 detection methods available in literature in multiple dimensions such as LOD, dynamic range, and clinical validation. Finally, we expanded the clinical validation study and improved the statistical analysis to clinical results.

We now believe that the revised manuscript will resonate with a board audience who is interested in building diagnostic technologies for various diseases.

Minor Comments:

1. A lot of work was done to compare the method to existing/alternative methodologies. It would benefit the reader to include a table comparing the methods either in the main text or supplemental (or to include these data in table S1).

RE: We thank the Reviewer for this great suggestion. We have included a comprehensive comparison to existing reports in Table S1 and Figure S15 of Supporting Information.

2. There are a few small typos throughout the manuscript that should be corrected, for example figures 3b and S6 are labeled "abumdance" rather than "abundance"

RE: We thank the Reviewer for pointing out these errors. We have corrected all typos and grammar errors in the revised manuscript.

- There are several instances in the text where the data is not presented objectively. The manuscript should be carefully reviewed to ensure the information is presented objectively. One example includes where the authors claim that their method demonstrates a 20-100 fold higher analytical sensitivity than existing methods yet the LOD reported is 0.5 pg/mL for the CAN assay and the LOD reported for a commercial ELISA kit: <https://www.abcam.com/hiv1-p24-elisa-kit-ab218268.html> is 1 pg/mL.

RE: We would like to clarify that the ELISA kit was used in *human sera* in the present study, while the reported 1 pg/mL LOD was achieved in different *work solutions (buffers)*. The following table from the vendor’s website shows more details about the LOD of the ELISA kit. In most cases, the LOD obtained in human serum is generally an order of magnitude higher than the LOD obtained in working buffer (*Angew. Chem. Int. Ed. 2018, 57, 11882-11887*).

The 10 pg/mL (0.41 pM) LOD reported in our manuscript for the ELISA kit was resulted from multiple tests using p24 spike human serum. This result has been extensively repeated and confirmed in our lab.

Sample Diluent Buffer	n=	Minimal Detectable Dose
Sample Diluent 50BS	16	1.5 pg/mL
Sample Diluent NS	32	1.3 pg/mL
1X Cell Extraction Buffer PTR	32	1.1 pg/mL

Table 1. The minimal detectable dose by the ELISA kit reported by Abcam.

- Please carefully examine Figure 2. I think there may be an error in the DNA structure shown in figure 2g, it looks like unmodified DNA, but is this trace showing alkyne modified DNA? Please clarify. If possible, please lower the red triangles down from the trace in 2e so it is easier to see the 2-level trend when comparing the three distinct traces.

RE: We thank the Reviewer for pointing out these errors. We have corrected the alkyne modified DNA structure and also have adjusted the triangle pointers in Figure 2 of the revised manuscript.

- Page 9 line 3: please check review the use of the word significant and its other forms without including the specific statistical values associated with any analysis performed. This would also apply to page 16 lines 6 and 7. For example, the authors may be required to include data as follows in the example “there was a significant increase in the positive versus control group ($F(2,27) = 1.4$, $p=0.1$)...”

RE: We thank the Reviewer for this suggestion. We have included specific statistical values when describing or comparing the abovementioned values as well as other values in the revised manuscript.

For comparison of different dwell time (Page 10), we have revised the statement as follows:

“However, its mean dwell time was significantly increased from 1.94 ± 0.50 ms to 19.34 ± 0.89 ms (Figure 2j, $P < 0.0001$, T_{DNA} versus T_{DNA-AA} by two-tailed unpaired Student t-test with ≥ 500 blockade events).”

For comparison of the CAN assay and ELISA in Figure 6, we have included the details as follows:

“Among all four groups, p24 was at similarly low levels in Negative and VL-0 groups ($F(11,33) = 17.97$, $p=0.98$), but was significantly higher in patients with detectable (LOD~30) viral load ($F(11,55) = 112.8$, $p < 0.05$), and high (Positive) viral load ($F(11,19) = 593.8$, $p < 0.0001$). Meanwhile, no significant difference was detected between p24 across Negative, VL-0, and LOD~30 groups measured by ELISA ($F(11,55) = 109.0$, $p=0.20$) (Figure 6d and e).”

6. The way figure 3 d is colored currently it looks like the legend only relates to the Fe ions.

RE: We thanks the Reviewer for this suggestion. We have revised the color and the legend in Figure 4a-b in the revised manuscript.

7. Figure 5 a is slightly confusing, some of the arrows seem in indicate contrary information, for example the clinical diagnosis: HIV panel points to both the HIV and non-HIV controls.

RE: We thank the reviewer for pointing out this issue. We have modified the participant disposition flow diagram in the revised manuscript for better clarity.

8. It is not clear why the COVID-19 patients were included? The group is not big enough to draw statistically valid comparison, and its presence is not discussed past the figure. Consider removing these data points until more samples are available for analysis.

RE: We appreciate the Reviewer’s comment, and have focused this study to HIV samples only.

9. On page 5 there are several statements that need to be corrected.

a. Line 1: The authors state that “By measurement and statistical analysis [...] via the frequency of the blockade events”. Maybe clarify that modified DNA molecules must be in $> \text{nM}$ for this to work or else it takes too long to capture a statistically significant number of molecules.

RE: We thank the Reviewer for this comment. This statement is for introducing the general mechanism of nanopore sensing, thus is not related to modified DNA molecules.

b. Line 3-4: The authors state that “However, due to [...] towards any analytes”, however there are examples of aHL nanopores distinguishing SNPs, 3’ to 5’ orientation, etc. This statement should be revised.

RE: We thank the Reviewer for this suggestion. However, the specificity mentioned here is from

a biosensing point of view, indicating that there is no recognition receptor in the pore to provide any specificity.

c. Line 7: the authors state “[...] not suitable for detecting electrically neutral molecules [...]”, yet many examples exist of such molecules being tested. Further not all proteins and antibodies are neutral, many have charges and thus a dipole moment. Please correct this statement and add appropriate references.

RE: We thank the Reviewer for this suggestion, and have revised the statement as follows (Page 4):

“However, due to its stochastic nature, it is difficult to specifically sense ultra-low abundance biomarkers mixed in interferent molecules in complex clinical specimens using a nanopore. In addition, sensitivity of direct detection of antigens and antibodies through electrophoresis-based nanopore sensing is limited, especially using biological nanopores with limited operating pH and fixed pore diameters.^{30,31}”

Appropriate references (*Small* 9, 750-759 (2013); *Nat. Commun.* 8, 1552 (2017)) were added in the revised manuscript.

10. On Page 8, the authors mention limiting their applied voltage to limit membrane capacitance. I do not understand this statement as their capacitance should depend on the area, thickness and dielectric properties of membrane but not on voltage.

RE: We thank the Reviewer for this comment. In our present approach, DPhPC was used to fabricate bilayer membranes for nanopore measurements. Typically, a higher applied potential is desired in sensing performance optimization (*Biophysical Journal*, 120 (9), 2021, 1537-1541). However, further increasing the applied potential may cause membrane rupture, which have been confirmed by other studies. (*Nat Commun* 12, 5811 (2021). DOI: 10.1038/s41467-021-26054-9;).

In this manuscript, the lipid membrane capacitance is 160-180 pF, corresponding to ~5nm thickness (Figure 2c). While the thickness or capacitance of the membrane is indeed not affected by voltage, the stability of the membrane will be weakened by high voltage (Figure S4). Consequently, an optimized voltage (160 mV) was found for DNA-AA@CB[6] probe detection to achieve maximum capture rate while maintaining membrane capacitance and stability (Figure 2f).

We have revised the corresponding statement to:

“In this study, a 160 mV operating voltage was employed to maintain membrane stability and capacitance (160-180 pF).”

11. On page 8, significant figures for blockages should match the error reported.

RE: We thank the Reviewer for pointing out this error. We have corrected this error in Figure 2 and corresponding descriptions (Page 10) in the revised manuscript.

Reviewers' Comments:

Reviewer #1:

Remarks to the Author:

The authors have done a thorough job of responding to all concerns. The revisions and additions have improved the manuscript, made the novelty and results clearer, and better justified the conclusions. I am satisfied with the response and suggest the paper be accepted for publication.

Reviewer #2:

Remarks to the Author:

Overall, the authors made substantial changes and performed a number of additional experiments to address the reviewers' comments. They certainly work very hard. I like that they have employed their nanopore assay on so many clinical samples. This is bringing the field in the right direction. Nonetheless, a number of basic comments remain unsatisfactorily addressed (capture rate calculation, error bars from single-molecule count, number of events detected, non-linear response in Figure 4g), which casts some doubts on the value of the conclusions: ability to precisely quantify the protein, so I cannot recommend publication until these are addressed.

Below are comments numbered following the authors' response letter.

1.1 The authors have sufficiently provided context and challenges based on their previous work to explain the motivation for this work, by incorporating two paragraphs. One that speaks to their previous work, and another that speaks to its limitations.

- Figure 1 schematically, and the modified caption is much improved. One question still remains from the schematic though, in the absence of the Cu⁺ ion does the DNA remain alkyne modified? If so the image of the DNA passing through the nanopore is misleading.
- In this section of added text: "the present CAN assay has two build-in amplifications by catalytic click reaction and biotin-avidin binding to achieve three orders of magnitude lower LOD," said to be on page 5-6, the statement that this method allows for an LOD that is 3 orders of magnitude is perhaps oversimplified. There may have been some contribution by the antibodies for each target that also contributed to the LOD. For example, if the currently used antibody pair has better affinity than the pair used in the previous work it would also contribute to a lower LOD. This statement should be revised to indicate that the amplifications contributed to a lower LOD but were not necessarily solely responsible.
- Also the use of the MinION imagery from ONT to depict nanopore detection in Figure 1 is misleading since the authors do not use an array of pores, nor the MinION system. This should be fixed to show a schematic of their setup

1.2 The authors have provided a clear explanation for the differences in the Cu⁺ and Cu²⁺ shown in the schematics and figures, and also clearly addressed this process in the text. This provides important context, and will be helpful to readers who are not familiar with the reaction.

1.3 The added experiments certainly clarify the necessary reaction conditions.

- One small comment for figure 3 f) the p24:0~145 pM should not cover the data. On 3g the data dot is labeled whereas it is not in figure 3f. Also, the R² is shown on the plot for 3g and is not for 3f. Also more ticks should be included on Figure 3f and 3g to be able to read the data off the figure.
- On page 11 line 199, it would be useful to have a reference, and for the authors to restate the clinically valid range of p24.
- One question that was not addressed was how the capture rate is analyzed. A comment is made that events were counted but that does not explain for the rate was extracted (for example, are the authors simply dividing the number of events by the recording time, and if so can they show

data that the event rate is constant during that recording, by showing that the cumulative number of events with time is linear or something). Furthermore, if the number of single-molecule counted is low then the error on the rate should be high and error calculation should be included.

- The peaks of interest in Figure S9 should be assigned in the caption of the figure for clarity.
- In response to the comment about distribution of sub-populations of DNA, DNA-AA and DNA-AA-CB[6], I understand what that the authors only count DNA-AA-CB[6], since there is variability in how the nanopore is able to detect or distinguish DNA or DNA-AA, but they fail to answer the simple question of how many single-molecules are counted and how is the capture rate of a sub-population extracted (unless it is as simple as dividing the number of multi-level events by the recording time, but then see comment above to convince the reader that this is a sound approach in these experiments) . Furthermore, the authors say "Due to the stochastic nature of nanopore sensing, signals of the same analyte (i.e. DNA or DNA-AA) are different from test to test. Thus, the distribution of DNA and DNA-AA subpopulations cannot be quantified by conventional statistics of nanopore signal either". This seem like an inaccurate statement; I would suggest that the authors more specifically acknowledge what part of the sensing system is deficient – are they limited by translocation times (too fast for their limited bandwidth)? signal is too similar? Analysis challenge?

2. On Figure 4g. I regret to keep pushing on this, but when a function appears linear on a log-log scale it indicates a relationship of the form $y=ax^n$, which is generally nonlinear unless $n=1$. In this case n (the slope of the line when fitted after log-transforming both x and y) appears to not equal to 1, and so the relationship is not linear. The authors should put the fit parameters in the caption and address this. For route I, the authors also fit on three points, excluding 2 out of 5. Therefore, statements like on page 14: "Excellent linear correlations between multi-level event frequencies and p24 concentrations were observed in both assays, respectively (route I $R^2 = 0.97$, route II $R^2 = 0.99$)." just do not make sense – clearly the authors see 1 log of event rate change for >3 logs of concentration change. This is not linear.

3. The authors have addressed some of the comments raised and performed additional experiments to improve their precision, which is commendable, but some comments are not addressed correctly. In particular, they do not directly discuss how many single-molecules are used to extract the capture rate (data used to quantify p24 concentration) nor how the capture is calculated (see point above). The number of single molecules should be indicated in the caption of Figure 4g (and in other relevant places – this is the norm in nanopore studies). From the partial answers of the authors assuming a 5 min recording at a capture of a few events per minutes, they only count 10 to 50 (multi-level) events to calculate event rates in Figure 4g. The authors should add a discussion of how their precision (error bar) is affected by this relatively low count. The authors say that "additional parallel experiments for each data point to improve the precision of data." How many experiments? This information should be included in the caption and the data shown in the SI. Prior to this the authors say that the results were obtained on 3 different pores. If the data needs to be averaged over 3 pores to reach a level of precision to reliably distinguish an order of magnitude change in p24 concentration how many pores were used for the clinical validation. There should be error bars in Figure 6a.

4. The authors do say they are confident about their LOD value, but I feel that LOD discussions should be revised based on the above.

5. Figure 6a should have error bars on the p24 concentration measurement (a table in the SI indicated number of multi-level counted per sample, as well as the total even count and the measurement time should be included). The grey, green and yellow circles (used as labels?) are complicating the reading of Figure 6a. This should be changed. The test statistics for the one-way ANOVA with post-hoc Tukey test should be reported in the caption.

All minor comments have been addressed well, except for point 9 a) and 9 b) where some additional clarification/context may be required in the manuscript.

Additional general comments.

1) Figure 3 b) there are no units for the mass

- 2) For the ^1H NMR, please stretch out the traces so they are not overlapping, or provide individual spectra in the SI for comparison
- 3) The figures tend to be very busy, especially with the insets. The authors may consider breaking the figures up or adding more into the supplemental for clarity.

Response to Reviewers' Comments on NCOMMS-21-13368B-Z

We greatly appreciate all the reviewers for their constructive and thoughtful comments. Below is a point-by-point response to each comment from the reviewers (under "RE:" in blue). Text added into the manuscript is shown in orange.

Response to Reviewer #1

The authors have done a thorough job of responding to all concerns. The revisions and additions have improved the manuscript, made the novelty and results clearer, and better justified the conclusions. I am satisfied with the response and suggest the paper be accepted for publication.

RE: We thank the Reviewer for all the thoughtful comments, which have helped to strengthen our work significantly.

Response to Reviewer #2

Overall, the authors made substantial changes and performed a number of additional experiments to address the reviewers' comments. They certainly work very hard. I like that they have employed their nanopore assay on so many clinical samples. This is bringing the field in the right direction. Nonetheless, a number of basic comments remain unsatisfactorily addressed (capture rate calculation, error bars from single-molecule count, number of events detected, non-linear response in Figure 4g), which casts some doubts on the value of the conclusions: ability to precisely quantify the protein, so I cannot recommend publication until these are addressed.

RE: We sincerely appreciate the Reviewer for commenting: "This is bringing the field in the right direction." Indeed, this is our exact motivation for starting this work: translation of nanopore-based single molecule sensing from bench to bedside. We strongly believe that the nanopore technology will do much more than its current powerful capacity in sequencing.

Inspired by the Reviewer's comments, we have added results and detailed discussions on: capture rate calculation; standard deviation (error bars) for all clinical samples from more single molecule counts; total numbers of events detected; the non-linear response in standard curves; and limitations of the current assay. Meanwhile, we have also addressed all the minor issues.

Again, we thank the Reviewers for their support and criticism to make our work better, and hope that our revisions and responses provided below sufficiently address their concerns.

Major Comments:

1. The authors have sufficiently provided context and challenges based on their previous work to

explain the motivation for this work, by incorporating two paragraphs. One that speaks to their previous work, and another that speaks to its limitations.

RE: We thank the Reviewer for acknowledging this improvement.

1.1 Figure 1 schematically, and the modified caption is much improved. One question still remains from the schematic though, in the absence of the Cu^+ ion does the DNA remain alkyne modified? If so the image of the DNA passing through the nanopore is misleading.

RE: We thank the Reviewer for pointing out this error. The DNA remains alkyne modified in absence of the Cu^+ ions. We have revised Figure 1 (now Scheme 1) accordingly:

Scheme 1. Illustration of the click chemistry amplified nanopore (CAN) assay workflow for quantification of HIV-1 p24 antigen in human serum: (1) blood samples were collected into serum separator tubes, and sera were collected after centrifugation and clot formation; (2) serum samples were incubated with capture antibody-modified magnetic beads (MBs) and detection antibody-modified copper oxide nanoparticles (CuONPs) to form a sandwich structure between MBs, enriched p24 antigens and CuONPs; (3) sandwich complexes were magnetically separated, and Cu^+ ions were released from them under acidic conditions; (4) DNA probes were formed by a Cu^+ ion catalyzed click reaction and a modification with a pair of host-guest molecules; (5) finally, DNA probes were collected and subjected to single-channel recordings using an α -hemolysin (α -HL) nanopore for p24 antigen quantification. The illustration is not drawn to scale. (Created with BioRender.com)

1.2 In this section of added text: “the present CAN assay has two build-in amplifications by catalytic click reaction and biotin-avidin binding to achieve three orders of magnitude lower LOD,” said to be on page 5-6, the statement that this method allows for an LOD that is 3 orders of magnitude is perhaps oversimplified. There may have been some contribution by the antibodies for each target that also contributed to the LOD. For example, if the currently used antibody pair has better affinity than the pair used in the previous work it would also contribute to a lower LOD. This statement should be revised to indicate that the amplifications contributed to a lower LOD but were not

necessarily solely responsible.

RE: We thank the Reviewer for this comment. We have revised the statement to (Page 6):

“the present CAN assay has two build-in amplifications by catalytic click reaction and biotin-avidin binding that contribute to a lower LOD to allow early detection of antigens in sera.”

1.3 Also, the use of the MinION imagery from ONT to depict nanopore detection in Figure 1 is misleading since the authors do not use an array of pores, nor the MinION system. This should be fixed to show a schematic of their setup.

RE: We thank the Reviewer for noticing this detail. We have revised the nanopore schematic in Scheme 1 to reflect our setup.

2. The authors have provided a clear explanation for the differences in the Cu^+ and Cu^{2+} shown in the schematics and figures, and also clearly addressed this process in the text. This provides important context, and will be helpful to readers who are not familiar with the reaction.

RE: We thank the Reviewer for acknowledging this improvement.

3. The added experiments certainly clarify the necessary reaction conditions.

RE: We thank the Reviewer for acknowledging this improvement.

3.1 One small comment for figure 3 f) the p24:0~145 pM should not cover the data. On 3g the data dot is labeled whereas it is not in figure 3f. Also, the R^2 is shown on the plot for 3g and is not for 3f. Also more ticks should be included on Figure 3f and 3g to be able to read the data off the figure.

RE: We thank the Reviewer for finding these detailed errors. We have revised Figure 3f-g (now Figure 2f-g) accordingly.

Figure 2. f: Reaction efficiency as a function of Cu^+ ion concentrations under 4 h reaction time with linear fit between 0 and 1 mM Cu^+ ions. Inset shows the theoretical p24 antigen concentration range covered. **g:** Correlation between multi-level signal frequency and Cu^+ ion concentration. Inset shows the correlation in

higher range up to 10 mM. Data represents mean \pm SD of three replicates ($N > 1000$ for each data point). Solid line indicates linear regression. Shadow indicates limits of 95% confidence interval.

3.2 On page 11 line 199, it would be useful to have a reference, and for the authors to restate the clinically valid range of p24.

RE: We appreciate the Reviewer for this suggestion. We have added the following related references and restated the clinical p24 range on page 12.

“...to cover quantification needs in most clinical scenarios (0.42-33.3 pM).”

(48) Anderson, A. M., Tyor, W. R., Mulligan, M. J., Waldrop-Valverde, D., Lennox, J. L., & Letendre, S. L. (2018). Measurement of human immunodeficiency virus p24 antigen in human cerebrospinal fluid with digital enzyme-linked immunosorbent assay and association with decreased neuropsychological performance. *Clinical Infectious Diseases*, 67(1), 137-140.

(49) Sutthent, R., Gaudart, N., Chokpaibulkit, K., Tanliang, N., Kanoksinsombath, C., & Chaisilwatana, P. (2003). p24 antigen detection assay modified with a booster step for diagnosis and monitoring of human immunodeficiency virus type 1 infection. *Journal of clinical microbiology*, 41(3), 1016-1022.

(50) Rissin, D. M., Kan, C. W., Campbell, T. G., Howes, S. C., Fournier, D. R., Song, L., ... & Duffy, D. C. (2010). Single-molecule enzyme-linked immunosorbent assay detects serum proteins at subfemtomolar concentrations. *Nature biotechnology*, 28(6), 595-599.

3.3 One question that was not addressed was how the capture rate is analyzed. A comment is made that events were counted but that does not explain for the rate was extracted (for example, are the authors simply dividing the number of events by the recording time, and if so, can they show data that the event rate is constant during that recording, by showing that the cumulative number of events with time is linear or something). Furthermore, if the number of single-molecule counted is low then the error on the rate should be high and error calculation should be included.

RE: We sincerely appreciate the Reviewer's constructive criticism and suggestions. In our current assay, the multi-level signature events of DNA-AA-CB[6] probes were counted over time and the capture rate was calculated by dividing the number of events over the valid recording time. This is supported by the following three experimental observations: (1) cumulative number of DNA-AA@CB[6] events increase linearly with the same rate over different recording time (Figure 1i); (2) cumulative number of DNA-AA@CB[6] events increase linearly with different Cu^+ ion concentrations, the event rate is positively correlated with Cu^+ ion concentrations (Figure 1j); (3) the DNA-AA@CB[6] capture rate is consistent among different nanopores (Figure 1l).

Figure 1. i-j: Cumulative numbers of multi-level events acquired using different independent nanopores (i) over different recording time and (j) with different Cu ion concentrations (total events $N > 500 \text{ min}^{-1}$). l: Capture rates of multi-level events recorded under 160 mV *trans* potential using five independent nanopores.

We have added the direct discussion of capture rate in the revised manuscript as follows (Pages 10-11):

“In the capture rate study, standard DNA-AA@CB[6] probe samples were tested using independent nanopores. Events induced by DNA-AA@CB[6] were extracted by observing their multi-level and oscillation characteristics and counted. Cumulative counting of the multi-level signature events shows the same increase rate of event numbers over different recording times (1, 2, 3, 5 minutes) and across four different pores for the same sample (Figure 1i), which allows capture rate calculation by dividing the event number over the recording time. While the total single-molecule event rate (alkyne modified DNA, DNA-AA, and DNA-AA@CB[6]) remains consistent at $>500 \text{ min}^{-1}$, positive correlation was observed between the capture rate of multi-level events and the Cu ion concentration used to form DNA-AA. The Cu ion concentrations tested (0.1, 0.5, 1 mM) were anticipated to cover most clinical samples (Figure 1j).” ...

... In this study, 160 mV operating voltage and 2-minute recording time was employed to optimize the sample-to-answer time while maintaining membrane stability as well as consistent capacitance (160-180 pF) and interpore capture rate (Figure 1l).

From these results, we can see that the cumulative number of multi-level events increases steadily from 1 minute to 5 minutes. Therefore a 2-minute recording time employed in our clinical tests can yield the same capture rate results as longer recording. Although the multi-level events rate is relatively low ($30\text{-}40 \text{ min}^{-1}$), the total single-molecule counted is over 500 per minute.

In addition, we have also included the total single-molecule counts (N) in figure captions where applicable and error calculations for all the clinical samples.

3.4 The peaks of interest in Figure S9 should be assigned in the caption of the figure for clarity.

RE: We thank the Reviewer for this suggestion. We have added the peak information in the caption of Figure S9 (Now Figure S10) as follows:

Figure S10. b: ^1H NMR (400 MHz) spectra of DNA-AA, CB[6] and DNA-AA@CB[6]. The binding interaction between DNA-AA and CB[6] complex can be deduced via the chemical shift of protons of the CB[6] molecule from methylene (CH_2) groups (5.68 ppm and 5.43 ppm) and tertiary C–H groups (5.64 ppm).^{5,6}

3.5 In response to the comment about distribution of sub-populations of DNA, DNA-AA and DNA-AA-CB[6], I understand what that the authors only count DNA-AA-CB[6], since there is variability in how the nanopore is able to detect or distinguish DNA or DNA-AA, but they fail to answer the simple question of how many single-molecules are counted and how is the capture rate of a sub-population extracted (unless it is as simple as dividing the number of multi-level events by the recording time, but then see comment above to convince the reader that this is a sound approach in these experiments). Furthermore, the authors say “Due to the stochastic nature of nanopore sensing, signals of the same analyte (i.e. DNA or DNA-AA) are different from test to test. Thus, the distribution of DNA and DNA-AA subpopulations cannot be quantified by conventional statistics of nanopore signal either”. This seem like an inaccurate statement; I would suggest that the authors more specifically acknowledge what part of the sensing system is deficient – are they limited by translocation times (too fast for their limited bandwidth)? signal is too similar? Analysis challenge?

RE: We thank the Reviewer for this comment. Our current study adopts a unified calculation method for the capture rate of DNA-AA@CB[6]: dividing the number of multi-level events over the recording time. This calculation method is statistically sound as demonstrated in the answer to comment 3.3, and was further validated using p24 spiked standard serum samples and clinical serum samples from HIV patients. For better clarification, here we briefly describe the workflow of the capture rate calculation for DNA-AA@CB[6]: over a >2 minutes valid recording time (excluding unstable baseline, complete blocking, etc.), we acquire a total of >1000 single molecule events (DNA, DNA-AA, DNA-AA@CB[6]). Events induced by DNA-AA@CB[6] were extracted by observing their multi-level and oscillation characteristics and counted. Then the capture rate (frequency) was calculated by dividing the multi-level event number over the valid recording time.

For DNA and DNA-AA, we did not calculate their capture rates nor try to distinguish between them. Their signals were excluded from the calculation. We also thank the Reviewer for pointing out our inaccurate statement, we have revised this point in the as follows (page 12):

“While the Cu^+ concentration dependency of DNA-AA formation was clearly characterized using HPLC (Figure S11a) and MS, nanopore measurements, on the other hand, cannot completely separate subpopulations of alkyne modified DNA and DNA-AA translocation events due to signal similarity (Figure S11b).”

We have also added single molecule event numbers in Figure S11b.

4. On Figure 4g. I regret to keep pushing on this, but when a function appears linear on a log-log scale it indicates a relationship of the form $y=ax^n$, which is generally nonlinear unless $n=1$. In this case n (the slope of the line when fitted after log-transforming both x and y) appears to not equal to 1, and so the relationship is not linear. The authors should put the fit parameters in the caption and address this. For route I, the authors also fit on three points, excluding 2 out of 5. Therefore, statements like on page 14: “Excellent linear correlations between multi-level event frequencies and p24 concentrations were observed in both assays, respectively (route I $R^2 = 0.97$, route II $R^2 = 0.99$).” just do not make sense – clearly the authors see 1 log of event rate change for >3 logs of concentration change. This is not linear.

RE: We sincerely appreciate the Reviewer’s constructive criticism and explanation, and apologize for our misunderstanding and misleading statements in the previous manuscript. We now have clarified this point (page 14-15) and revised all other related statements throughout the manuscript. Figure 4g (now Figure 3g) was also revised with fitted constants in the captions.

“A positive correlation between multi-level event frequencies and p24 concentrations were observed in both assays (inset of Figure 3g, wide range).”

Figure 3g. g: Correlations between multi-level signal frequency and p24 concentration in human serum within the range of 0-46.7 pM (0-1000 pg/mL, final concentration: 0, 0.5, 1, 10, 100, 1000 pg/mL) processed using route I and route II assays, respectively. Insets show the correlations within lower range (0-50 fM) and wide range (0-40 pM). Data represents mean \pm SD of five parallel experiments ($N > 2000$)

for each data point, detailed information is shown in Table S1). Solid line indicates linear regression with slopes of 0.315 ± 0.045 (route I, $R^2 = 0.97$, equation: $y = 0.590x^{0.315}$) and 0.261 ± 0.004 (route II, $R^2 = 0.99$, equation: $y = 0.818x^{0.261}$). Shadow indicates limits of 95% confidence interval.

5. The authors have addressed some of the comments raised and performed additional experiments to improve their precision, which is commendable, but some comments are not addressed correctly. In particular, they do not directly discuss how many single-molecules are used to extract the capture rate (data used to quantify p24 concentration) nor how the capture is calculated (see point above). The number of single molecules should be indicated in the caption of Figure 4g (and in other relevant places – this is the norm in nanopore studies). From the partial answers of the authors assuming a 5 min recording at a capture of a few events per minutes, they only count 10 to 50 (multi-level) events to calculate event rates in Figure 4g. The authors should add a discussion of how their precision (error bar) is affected by this relatively low count. The authors say that “additional parallel experiments for each data point to improve the precision of data.” How many experiments? This information should be included in the caption and the data shown in the SI. Prior to this the authors say that the results were obtained on 3 different pores. If the data needs to be averaged over 3 pores to reach a level of precision to reliably distinguish an order of magnitude change in p24 concentration how many pores were used for the clinical validation. There should be error bars in Figure 6a.

RE: We thank the Reviewer for these comments and suggestions. We have broken down this section to several issues and responded point by point below.

- 5.1 In particular, they do not directly discuss how many single-molecules are used to extract the capture rate (data used to quantify p24 concentration) nor how the capture is calculated (see point above).

RE: We thank the Reviewer for this suggestion, and have added a discussion about capture rate calculation (See point 3.3). We have also added the numbers of single molecule events (N) in several figure captions throughout the manuscript where applicable.

- 5.2 The number of single molecules should be indicated in the caption of Figure 4g (and in other relevant places – this is the norm in nanopore studies). From the partial answers of the authors assuming a 5 min recording at a capture of a few events per minutes, they only count 10 to 50 (multi-level) events to calculate event rates in Figure 4g.

RE: We thank the Reviewer for this suggestion. We have added in Figure 4g (now Figure 3g) caption that the total single molecule event number for each data point is $N > 2000$. This indicates that, for each p24 concentration, we recorded a total of over 2000 single molecule translocation events from 5 repeated experiments. Among these events, multi-level signals were counted and divided by recording time to calculate the capture rate.

We have also summarized the multi-level event capture rate from each pore, the total recording time, and the total multi-level events number counted for each p24 concentration point of both routes in Table S1:

Table S1. Capture rate of multi-level signal obtained from parallel experiments for various p24 concentration in human serum.

Route	p24 pg/mL	p24 pM	Capture Rate (min ⁻¹)					Recording Time (s)	Counted Events
			Pore 1	Pore 2	Pore 3	Pore 4	Pore 5		
Route II	0	0	0	0.83	0.6	0.6	0	281	2
	0.5	0.021	2.65	2.35	2.26	2.2	2.1	579.2	22
	1	0.042	2.78	3.79	3.03	2.78	2.74	926	47
	10	0.417	4.2	5.43	5.78	5.88	5.71	1252	113
	100	4.167	9.67	10.13	10.1	9.86	8.58	781	126
	1000	41.667	17.25	17	18.9	15.8	17.24	741.4	213
Route I	0	0	0	0	0.79	0.88	0.6	452.2	3
	0.5	0.021	0	0.69	0	0	1.29	356	2
	1	0.042	0	0.66	1.45	0	0	588.7	4
	10	0.417	2.77	3.03	2.96	2.7	3.57	427	22
	100	4.167	7.22	5.92	6.33	6	7.4	693.5	76
	1000	41.667	10.52	12	13.8	9.77	11.11	540.5	103

Similarly, we have added in Figure 6a (now Figure 4a) caption that the total single molecule event number for each clinical sample is $N > 1000$. We have also summarized the multi-level event capture rate from each pore, the total recording time, and the calculated p24 concentration for each clinical sample in Table S3.

5.3 The authors should add a discussion of how their precision (error bar) is affected by this relatively low count.

RE: We thank the Reviewer for this suggestion, and have added the discussion of how the precision is affected by the relatively low count as follows (pages 15 and 22):

“However, in comparison with route I in which accuracy and reproducibility were greatly affected by low signal frequency, the biotin-streptavidin amplification strategy (route II) greatly improved detection accuracy and reproducibility, especially in the low p24 concentration range (0~42 fM), to achieve approximately 20-fold lower LOD than that of route I assay (inset of Figure 3g, lower range) while reliably quantify changes in p24 concentration.”

“Secondly, elevated standard deviations were observed in clinical results as normally seen in previous clinical diagnostics studies, especially in the lower p24 range, which may cause false calling of results close to LOD. This is likely due to the relatively low event counts in these samples and the higher complexity of interferents in HIV patients’ sera comparing to standard spiked sera samples. Strategies for improving probe DNA capture rate such as increasing the driving force using higher voltage and probes with higher charges, increasing the p24 to probe DNA ratio using advanced particle materials and linkers, as well as reducing the sample complexity by separating common blood interferents are currently under study.”

5.4 The authors say that “additional parallel experiments for each data point to improve the precision of data.” How many experiments? This information should be included in the caption and the data shown in the SI.

RE: We thank the Reviewer for this comment. There are five parallel experiments with different nanopores for each data point in the standard curves and three parallel experiments with different nanopores for each clinical sample. We have added this information in the caption of Figure 3g and Figure 4a. The detailed data from each parallel experiment is shown in the Supporting Information file (Table S1 and S3)

5.5 Prior to this the authors say that the results were obtained on 3 different pores. If the data needs to be averaged over 3 pores to reach a level of precision to reliably distinguish an order of magnitude change in p24 concentration how many pores were used for the clinical validation. There should be error bars in Figure 6a.

RE: We thank the Reviewer for this suggestion. For each clinical sample, three different pores were used. We now have included error bars for all clinical samples in Figure 4a.

Figure 4a: Clinical validation in a pilot cohort. **a:** Quantitative measurements of p24 in clinical samples using the CAN assay, with the clinical diagnosis of each patient. Bar graph data represents mean \pm SD of three parallel measurements (total events $N > 1000$ for each sample, detailed information shown in Table S3). Each red diamond represents the result from one measurement.

- The authors do say they are confident about their LOD value, but I feel that LOD discussions should be revised based on the above.

RE: We thank the Reviewer for this comment. From all previous answers, we have demonstrated that the multi-level signal capture rate calculation can accurately reflect the p24 concentration of a sample. In Figure 3g inset for lower range, we observed that, in route II, the data point (mean \pm SD) of 20.8 fM can be clearly differentiated from the data point (mean \pm SD) of the blank sample. In Figure S13, we have also shown that multi-level signals can be observed in the raw data of 20.8

fM (0.5 pg/mL) sample but not in the blank sample. In addition, we also calculated the LOD using a well-established method (EP17 protocol) in clinical chemistry (see page 16). The calculation result (18.3 fM) confirmed our observations.

However, we did observe elevated standard deviations in some clinical results. Although this is normally seen in clinical results due to the fact that patients' sera normally contain more interferents than standard serum samples, the elevated standard deviations may still cause inaccuracy and inconsistency in low concentration results (low multi-level event counts). We have added a discussion to this point as follows (page 22).

“Secondly, elevated standard deviations were observed in clinical results as normally seen in previous clinical diagnostics studies,^{79,80} especially in the lower p24 range, which may cause false calling of results close to LOD. This is likely due to the relatively low event counts in these samples and the higher complexity of interferents in HIV patients' sera comparing to standard spiked sera samples. Strategies for improving probe DNA capture rate such as increasing the driving force using higher voltage and probes with higher charges, increasing the p24 to probe DNA ratio using advanced particle materials and linkers, as well as reducing the sample complexity by separating common blood interferents are currently under study.”

7. Figure 6a should have error bars on the p24 concentration measurement (a table in the SI indicated number of multi-levels counted per sample, as well as the total even count and the measurement time should be included). The grey, green and yellow circles (used as labels?) are complicating the reading of Figure 6a. This should be changed. The test statistics for the one-way ANOVA with post-hoc Tukey test should be reported in the caption.

RE: We thank the Reviewer for this suggestion. We have added Table S3 in the Supporting Information file to show the multi-level signal capture rate from each nanopore for each clinical sample and the total measurement time.

According to the Reviewer's suggestion, we have improved the presentation of Figure 6 (now Figure 4) for better clarity and included the test statistics of the one-way ANOVA with post-hoc Tukey test in the caption of Figure 4.

Minor Comments:

1. All minor comments have been addressed well, except for point 9 a) and 9 b) where some additional clarification/context may be required in the manuscript.

(The following comments are from the previous review)

Point 9a: The authors state that “By measurement and statistical analysis [...] via the frequency of the blockade events”. Maybe clarify that modified DNA molecules must be in > nM for this to work or else it takes too long to capture a statistically significant number of molecules.

Point 9b: The authors state that “However, due to [...] towards any analytes”, however there are examples of aHL nanopores distinguishing SNPs, 3’ to 5’ orientation, etc. This statement should be revised.

RE: We thank the Reviewer for the additional comments, for point 9a, we have revised the corresponding statement to (Page 4):

“By measurement and statistical analysis of ionic current blockades produced by translocations of individual target analytes through a single nanopore under an applied electrical potential, the concentration of an analyte can be obtained *via* the frequency of blockade events.²⁴⁻²⁷ In general, concentration of analyte, such as DNA, should be at least at nanomolar level to capture a statistically significant number of translocation events in a reasonable measurement time.²⁸”

For point 9b, we have revised the corresponding statement to (Page 4):

“However, due to its stochastic nature, it is difficult to specifically sense ultra-low abundance biomarkers mixed with interferent molecules in complex clinical specimens using a nanopore without any recognition receptors.³¹”

2. Figure 3 b) there are no units for the mass

RE: We thank the Reviewer for pointing out this error and have added the units in the revised Figure 2b.

3. For the ¹HNMR, please stretch out the traces so they are not overlapping, or provide individual spectra in the SI for comparison.

RE: We thank the Reviewer for this suggestion and have revised all the ¹HNMR related figures including Figure 3e and Figure S10b.

4. The figures tend to be very busy, especially with the insets. The authors may consider breaking the figures up or adding more into the supplemental for clarity.

RE: We thank the Reviewer for this suggestion. We have adjusted the layout of all the Figures and moved several results to the Supporting Information file.